# Consistent Estimation for PCA and Sparse Regression with Oblivious Outliers

**Tommaso d'Orsi**
ETH Zürich
Zürich, Switzerland

**Chih-Hung Liu**
ETH Zürich
Zürich, Switzerland

**Rajai Nasser**
ETH Zürich
Zürich, Switzerland

**Gleb Novikov**
ETH Zürich
Zürich, Switzerland

**David Steurer**
ETH Zürich
Zürich, Switzerland

**Stefan Tiegel**
ETH Zürich
Zürich, Switzerland

## Abstract

We develop machinery to design efficiently computable and *consistent* estimators, achieving estimation error approaching zero as the number of observations grows, when facing an oblivious adversary that may corrupt responses in all but an $\alpha$ fraction of the samples. As concrete examples, we investigate two problems: sparse regression and principal component analysis (PCA). For sparse regression, we achieve consistency for optimal sample size $n \gtrsim (k \log d)/\alpha^2$ and optimal error rate $O(\sqrt{(k \log d)/(n \cdot \alpha^2)})$ where $n$ is the number of observations, $d$ is the number of dimensions and $k$ is the sparsity of the parameter vector, allowing the fraction of inliers to be inverse-polynomial in the number of samples. Prior to this work, no estimator was known to be consistent when the fraction of inliers $\alpha$ is $o(1/\log \log n)$, even for (non-spherical) Gaussian design matrices. Results holding under weak design assumptions and in the presence of such general noise have only been shown in dense setting (i.e., general linear regression) very recently by d'Orsi et al. (dNS21). In the context of PCA, we attain optimal error guarantees under broad spikiness assumptions on the parameter matrix (usually used in matrix completion). Previous works could obtain non-trivial guarantees only under the assumptions that the measurement noise corresponding to the inliers is polynomially small in $n$ (e.g., Gaussian with variance $1/n^2$).

To devise our estimators, we equip the Huber loss with non-smooth regularizers such as the $\ell_1$ norm or the nuclear norm, and extend d'Orsi et al.'s approach (dNS21) in a novel way to analyze the loss function. Our machinery appears to be easily applicable to a wide range of estimation problems. We complement these algorithmic results with statistical lower bounds showing that the fraction of inliers that our PCA estimator can deal with is optimal up to a constant factor.

## 1 Introduction

Estimating information from structured data is a central theme in statistics that by now has found applications in a wide array of disciplines. On a high level, a typical assumption in an estimation problem is the existence of a –known *a priori*– family of probability distributions $\mathcal{P} := \left\{ \mathbb{P}_\beta \mid \beta \in \Omega \right\}$ over some space $\mathcal{Z}$ that are each indexed by some parameter $\beta \in \Omega$. We then observe a collection

35th Conference on Neural Information Processing Systems (NeurIPS 2021).

of $n$ independent observations $\mathbf{Z} = (\mathbf{Z}_1, \ldots, \mathbf{Z}_n)$ drawn from an unknown probability distribution $\mathbb{P}_{\beta^*} \in \mathcal{P}$. The goal is to (approximately) recover the *hidden* parameter $\beta^*$.[1]

Oftentimes, real-world data may contain skewed, imprecise or corrupted measurements. Hence, a desirable property for an estimator is to be robust to significant, possibly malicious, noise perturbations on the given observations. Indeed, in the last two decades, a large body of work has been developed on designing robust algorithms (e.g see (BGN09; BBC11; MMYSB19)). However, proving strong guarantees often either demands strong assumptions on the noise model or requires that the fraction of perturbed observations is small. More concretely, when we allow the noise to be chosen adaptively, i.e., chosen dependently on the observations and hidden parameters, a common theme is that *consistent* estimators – estimators whose error tends to zero as the number of observations grows – can be attained only when the fraction of outliers is small.

In order to make vanishing error possible in the presence of large fractions of outliers, it is necessary to consider weaker adversary models that are *oblivious* to the underlying structured data. In recent years, a flurry of works have investigated oblivious noise models (CLMW11; ZLW+10; TJSO14; BJKK17; SZF18; SBRJ19; PJL21; dNS21). These results however are tailor-made to the specific models and problems. To overcome this limitation, in this paper we aim to provide a simple blueprint to design provably robust estimators under minimalistic noise assumptions for a large class of estimation problems. As a testbed for our blueprint, we investigate two well-studied problems:

*Principal component analysis (PCA)*: Given a matrix $\mathbf{Y} := L^* + \mathbf{N}$ where $L^* \in \mathbb{R}^{n \times n}$ is an unknown parameter matrix and $\mathbf{N}$ is an $n$-by-$n$ *random* noise matrix, the goal is to find an estimator $\hat{L}$ for $L^*$ which is as close as possible to $L^*$ in Frobenius norm.

*Sparse regression*: Given observations $(X_1, \mathbf{y}_1), \ldots, (X_n, \mathbf{y}_n)$ following the linear model $\mathbf{y}_i = \langle X_i, \beta^* \rangle + \boldsymbol{\eta}_i$ where $X_i \in \mathbb{R}^d$, $\beta^* \in \mathbb{R}^d$ is the $k$-sparse parameter vector of interest (by $k$-sparse we mean that it has at most $k$ nonzero entries) and $\boldsymbol{\eta}_1, \ldots \boldsymbol{\eta}_n$ is noise, the goal is to find an estimator $\hat{\beta}$ for $\beta^*$ achieving small *squared prediction error* $\frac{1}{n}\|X(\hat{\beta} - \beta^*)\|^2$, where $X$ is the matrix whose rows are $X_1, \ldots, X_n$.[2]

**Principal component analysis.** A natural way to describe principal component analysis under oblivious perturbations is that of assuming the noise matrix $\mathbf{N}$ to be an $n$-by-$n$ matrix with a uniformly random set of $\alpha \cdot n^2$ entries bounded by some small $\zeta \geqslant 0$ in absolute value. In these settings, we may think of the $\alpha$ fraction of entries with small noise as the set of uncorrupted observations. Moreover, for $\zeta > 0$, the fact that even for uncorrupted observations the noise is non-zero allows us to capture both gross sparse errors and small entry-wise noise in the measurements at the same time (for example if $\zeta = 1$ then the model captures settings with additional standard Gaussian noise). Remarkably, for $\zeta = 0$, Candès et al.'s seminal work (CLMW11) provided an algorithm that exactly recovers $L^*$ even for a vanishing fraction of inliers, under *incoherence* conditions on the signal $L^*$. The result was slightly extended in (ZLW+10) where the authors provided an algorithm recovering $L^*$ up to squared error $O(\zeta^2 \cdot n^4)$ thus allowing polynomially small measurement noise $\zeta$, but still failing to capture settings where standard Gaussian measurement noise is added to the sparse noise. Even for simple signal matrices $L^*$, prior to this work, it remained an open question whether consistent estimators could be designed in presence of both oblivious noise and more reasonable measurement noise (e.g., standard Gaussian).

**Linear regression.** Similarly to the context of principal component analysis, a convenient model for oblivious adversarial corruptions is that of assuming the noise vector $\boldsymbol{\eta}$ to have a random set of $\alpha \cdot n$ entries bounded by $\zeta$ in absolute value (here all results of interest can be extended to any $\zeta > 0$ just by scaling, so we consider only the case $\zeta = 1$). This model trivially captures the classical settings with Gaussian noise (a Gaussian vector with variance $\sigma^2$ will have an $\alpha = \Theta(\frac{1}{\sigma})$ fraction of inliers) and, again, allows us to think of the $\alpha$-fraction of entries with small noise as the set of uncorrupted observations. Early works on consistent regression focused on the regime with Gaussian design $X_1, \ldots, X_n \sim N(0, \Sigma)$ and deterministic noise $\eta$.[3] (BJKK17) presented an estimator achieving error

---

[1]In this paper, we assume $\Omega \subseteq \mathbb{R}^d$ and $\mathcal{Z} \subseteq \mathbb{R}^D$ for some $d, D$, denote random variables in bold face, and hide absolute constant factors with the notation $O(\cdot), \Omega(\cdot), \gtrsim, \lesssim$ and logarithmic factors with $\tilde{O}(\cdot), \tilde{\Omega}(\cdot)$.

[2]Our analysis also works for the *parameter error* $\|\beta^* - \hat{\beta}\|$.

[3]For Gaussian design $X$ one may also consider a deterministic noise model. This noise model is subsumed by the random noise model discussed here (if $X$ is Gaussian). As shown in (dNS21), roughly speaking the

$\tilde{O}(d/(\alpha^2 \cdot n))$ for any $\alpha$ larger than a fixed constant. (SBRJ19) extended this result by achieving comparable error rates even for a vanishing fraction of inliers $\alpha \gtrsim 1/\log\log n$. Assuming $n \gtrsim d^2/\alpha^2$ (TJSO14) proved that the Huber-loss estimator (Hub64) achieves optimal error rate $O(d/\alpha^2 \cdot n)$ even for polynomially small fraction of inliers $\alpha \gtrsim \sqrt{d/n}$. This line of work culminated in (dNS21) which extended the result of (TJSO14) achieving the same guarantees with optimal sample complexity $n \gtrsim d/\alpha^2$. For Gaussian design, similar result as (dNS21) can be extracted from independent work (PF20). Furthermore, the authors of (dNS21) extended these guarantees to deterministic design matrices satisfying only a spreadness condition (trivially satisfied by sub-Gaussian design matrices).

Huber loss was also analyzed in context of linear regression in (SZF18; DT19; SF20; PJL21). These works studied models that are different from our model, and these results do not work in the case $\alpha = o(1)$. (SZF18) used assumptions on the moments of the noise, (DT19) and (SF20) studied the model with non-oblivious adversary, and the model from (PJL21) allows corruptions in covariates.

In the context of sparse regression, less is known. When the fraction of observations is constant $\alpha \geqslant \Omega(1)$, (NTN11) provided a first consistent estimator. Later (DT19) and (SF20) improved that result (assuming $\alpha \geqslant \Omega(1)$, but these results also work with non-oblivious adversary). In the case $\alpha = o(1)$, the algorithm of (SBRJ19) (for Gaussian designs) achieves nearly optimal error convergence $\tilde{O}\big((k\log^2 d)/\alpha^2 n\big)$, but requires $\alpha \gtrsim 1/\log\log n$. More recently, (dNS21) presented an algorithm for standard Gaussian design $X \sim N(0,1)^{n \times d}$, achieving the nearly optimal error convergence $\tilde{O}(k/(\alpha^2 \cdot n))$ for nearly optimal inliers fraction $\alpha \gtrsim \sqrt{(k \cdot \log^2 d)/n}$. Both (SBRJ19) and (dNS21) use an iterative process, and require a bound $\|\beta^*\| \leqslant d^{O(1)}$ (for larger $\|\beta^*\|$ these algorithms also work, but the fraction of inliers or the error convergence is worse). The algorithm from (dNS21) , however, heavily relies on the assumption that $X \sim N(0,1)^{n \times d}$ and appears unlikely to be generalizable to more general families of matrices (including non-spherical Gaussian designs).

**Our contribution.** We propose new machinery to design efficiently computable *consistent estimators* achieving optimal error rates and sample complexity against oblivious outliers. In particular, we extend the approach of (dNS21) to structured estimation problems, by finding a way to exploit adequately the structure therein. While consistent estimators have already been designed under more benign noise assumptions (e.g. the LASSO estimator for sparse linear regression under Gaussian noise), it was previously unclear how to exploit this structure in the setting of oblivious noise. One key consequence of our work is hence to demonstrate what minimal assumptions on the noise are sufficient to make effective recovery (in the sense above) possible. Concretely, we show

*Oblivious PCA*: Under mild assumptions on the noise matrix $N$ and common assumption on the parameter matrix $L^*$ –traditionally applied in the context of matrix completion (NW12)– we provide an algorithm that achieves optimal error guarantees.

*Sparse regression*: Under mild assumptions on the design matrix and the noise vector –similar to the ones used in (dNS21) for dense parameter vectors $\beta^*$ – we provide an algorithm that achieves optimal error guarantees and sample complexity.

For both problems, our analysis improves over the state-of-the-art and recovers the classical optimal guarantees, not only for Gaussian noise, but also under much less restrictive noise assumptions. At a high-level, we achieve the above results by equipping the Huber loss estimator with appropriate regularizers. Our techniques closely follow standard analyses for $M$-estimators, but crucially depart from them when dealing with the observations with large perturbations. Furthermore, our analysis appears to be mechanical and thus easily applicable to many different estimation problems.

## 2    Results

Our estimators are based on regularized versions of the *Huber loss*. The regularizer we choose depends on the underlying structure of the estimation problem: We use $\ell_1$ regularization to enforce sparsity in linear regression and nuclear norm regularization to enforce a low-rank structure in the context of PCA. More formally, the *Huber penalty* is defined as the function $f_h : \mathbb{R} \to \mathbb{R}_{\geqslant 0}$ such that

---

underlying reason is that the Gaussianity of $X$ allows one to obtain several other desirable properties "for free". For example, one could ensure that the noise vector is symmetric by randomly flipping the sign of each observation $(y_i, X_i)$, as the design matrix will still be Gaussian. See Section 2 for a more in-depth discussion.

$$f_h(t) := \begin{cases} \frac{1}{2}t^2 & \text{for } |t| \leqslant h \,, \\ h(|t| - \frac{h}{2}) & \text{otherwise.} \end{cases} \tag{2.1}$$

where $h > 0$ is a penalty parameter. For $X \in \mathbb{R}^D$, the *Huber loss* is defined as the function $F_h(X) := \sum_{i \in [D]} f_h(X_i)$. We will define the regularized versions in the following sections. For a matrix $A$, we use $\|A\|, \|A\|_{\text{nuc}}, \|A\|_{\text{F}}, \|A\|_{\max}$ to denote its spectral, nuclear, Frobenius, maximum[4] norms, respectively. For a vector $v$, we use $\|v\|$ and $\|v\|_1$ to denote its $\ell_2$ and $\ell_1$ norms.

## 2.1 Oblivious principal component analysis

For oblivious PCA, we provide guarantees for the following estimator ($\zeta$ will be defined shortly):

$$\hat{L} := \operatorname*{argmin}_{L \in \mathbb{R}^{n \times n}, \, \|L\|_{\max} \leqslant \rho/n} \left( F_h(Y - L) + 100\sqrt{n}\left(\zeta + \rho/n\right)\|L\|_{\text{nuc}} \right). \tag{2.2}$$

**Theorem 2.1.** *Let $L^* \in \mathbb{R}^{n \times n}$ be an unknown deterministic matrix and let $N$ be an $n$-by-$n$ random matrix with independent, symmetrically distributed (about zero) entries and $\alpha := \min_{i,j \in [n]} \mathbb{P}\{|N_{ij}| \leqslant \zeta\}$ for some $\zeta \geqslant 0$. Suppose that $\operatorname{rank}(L^*) = r$ and $\|L^*\|_{\max} \leqslant \rho/n$.*

*Then, with probability at least $1 - 2^{-n}$ over $N$, given $Y = L^* + N$, $\zeta$ and $\rho$, the estimator Eq. (2.2) with Huber parameter $h = \zeta + \rho/n$ satisfies*

$$\left\|\hat{L} - L^*\right\|_{\text{F}} \leqslant O\left(\frac{\sqrt{rn}}{\alpha}\right) \cdot (\zeta + \rho/n) \,.$$

We first compare the guarantees of Theorem 2.1 with the previous results on robust PCA (CLMW11; ZLW+10).[5] The first difference is that they require $L^*$ to satisfy certain incoherent conditions.[6] Concretely, they provide theoretical guarantees for $r \leqslant O\left(\mu^{-1}n(\log n)^{-2}\right)$, where $\mu$ is the incoherence parameter. In certain regimes, such a constraint strongly binds with the eigenvectors of $L^*$, restricting the set of admissible signal matrices. Using the different assumption $\|L^*\|_{\max} \leqslant \rho/n$ (commonly used for matrix completion, see (NW12)), we can obtain nontrivial guarantees (i.e. $\left\|\hat{L} - L^*\right\|_{\text{F}} / \|L^*\|_{\text{F}} \to 0$ as $n \to \infty$) even when the $\mu$-incoherence conditions are not satisfied for any $\mu \leqslant n/\log^2 n$, and hence the results (CLMW11; ZLW+10) cannot be applied. We remark that, without assuming incoherence, the dependence of the error on the maximal entry of $L^*$ is inherent (see Remark 3.2).

The second difference is that Theorem 2.1 provides a significantly better dependence on the magnitude $\zeta$ of the entry-wise measurement error. Specifically, in the settings of Theorem 2.1, (ZLW+10) showed that if $L^*$ satisfies the incoherence conditions, the error of their estimator is $O\left(n^2\zeta\right)$. If the entries of $N$ are standard Gaussian with probability $\alpha$ (and hence $\zeta \leqslant O(1)$), and the entries of $L^*$ are bounded by $O(1)$, then the error of our estimator is $O(\sqrt{rn}/\alpha)$, which is considerably better than $O(n^2)$ as in (ZLW+10). On the other hand, their error does not depend on the magnitude $\rho/n$ of the signal entries, so in the extreme regimes when the singular vectors of $L^*$ satisfy the incoherence conditions but $L^*$ has very large singular values (so that the magnitude of the entries of $L$ is significantly larger than $n$), their analysis provides better guarantees than Theorem 2.1.

As another observation to understand Theorem 2.1, notice that our robust PCA model also captures the classical matrix completion settings. In fact, any instance of matrix completion can be easily transformed into an instance of our PCA model: for the entries $(i, j)$ that we do not observe, we can set $N_{i,j}$ to some arbitrarily large value $\pm C(\rho, n) \gg \rho/n$, making the signal-to-noise ratio of the entry arbitrarily small. The observed (i.e. uncorrupted) $\Theta(\alpha \cdot n^2)$ entries may additionally be perturbed by Gaussian noise with variance $\Theta(\zeta^2)$. The error guarantees of the estimator in Theorem 2.1 is

---

[4]For $n \times m$ matrix $A$, $\|A\|_{\max} = \max_{i \in [n], j \in [m]} |A_{ij}|$.

[5]We remark that in (CLMW11) the authors showed that they can consider non-symmetric noise when the fraction of inliers is large $\alpha \geqslant 1/2$. However for smaller fraction of inliers their analysis requires the entries of the noise to be symmetric and independent, so for $\alpha < 1/2$, their assumptions are captured by Theorem 2.1.

[6]A rank-$r$ $n \times n$ dimensional matrix $M$ is $\mu$-*incoherent* if its *singular vector decomposition* $M := U\Sigma V^\top$ satisfies $\max_{i \in [n]} \|U^\top e_i\|^2 \leqslant \frac{\mu r}{n}$, $\max_{i \in [n]} \|V^\top e_i\|^2 \leqslant \frac{\mu r}{n}$ and $\|UV^\top\|_\infty \leqslant \frac{\sqrt{\mu r}}{n}$.

$O\big((\rho/n + \zeta)\sqrt{rn}/\alpha\big)$. Thus, the dependency on the parameters $\rho, n, \zeta$, and $r$ is the *same* as in matrix completion and the error is within a factor of $\Theta(\sqrt{1/\alpha})$ from the optimum for matrix completion. However, this worse dependency on $\alpha$ is intrinsic to the more general model considered and it turns out to be optimal (see Theorem 2.4). On a high level, the additional factor of $\Theta(\sqrt{1/\alpha})$ comes from the fact that in our PCA model we do not know which entries are corrupted. The main consequence of this phenomenon is that a condition of the form $\alpha \gtrsim \sqrt{r/n}$ appears *inherent* to achieve consistency. To get some intuition on why this condition is necessary, consider the Wigner model where we are given a matrix $Y = xx^\top + \sigma W$ for a flat vector $x \in \{\pm 1\}^n$ and a standard Gaussian matrix $W$. Note, that the entries of $W$ fit our noise model for $\zeta = 1, \rho/n = 1, r = 1$ and $\alpha = \Theta(1/\sigma)$. The spectral norm of $\sigma W$ concentrates around $2\sigma\sqrt{n}$ and thus it is information-theoretically impossible to approximately recover the vector $x$ for $\sigma = 1/\alpha = \omega(\sqrt{n})$ (see (PWBM16)).

## 2.2 Sparse regression

Our regression model considers a *fixed* design matrix $X \in \mathbb{R}^{n \times d}$ and observations $y := X\beta^* + \eta \in \mathbb{R}^n$ where $\beta^*$ is an *unknown* $k$-sparse parameter vector and $\eta$ is *random* noise with $\mathbb{P}(|\eta_i| \leq 1) \geq \alpha$ for all $i \in [n]$. Earlier works (BJKK17), (SBRJ19) focused on the setting that the design matrix consists of i.i.d. rows with Gaussian distribution $\mathcal{N}(0, \Sigma)$ and the noise is $\eta = \zeta + w$ where $\zeta$ is deterministic $(\alpha \cdot n)$-sparse vector and $w$ is subgaussian. As in (dNS21), our results for a fixed design and random noise can, in fact, extend to yield the same guarantees for this early setting (see Theorem 2.3). Hence, a key advantage of our results is that the design $X$ does not have to consist of Gaussian entries. Remarkably, we can handle arbitrary deterministic designs as long as they satisfy some mild conditions. Concretely, we make the following three assumptions, the first two of which are standard in the literature of sparse regression (e.g., see (Wai19), section 7.3):

1. For every column $X^i$ of $X$, $\|X^i\| \leq \sqrt{n}$.

2. *Restricted eigenvalue property (RE-property)*: For every vector $u \in \mathbb{R}^d$ such that[7] $\|u_{\text{supp}(\beta^*)}\|_1 \geq 0.1 \cdot \|u\|_1$, we have $\frac{1}{n}\|Xu\|^2 \geq \lambda \cdot \|u\|^2$ for some parameter $\lambda > 0$.

3. *Well-spreadness property*: For some (large enough) $m \in [n]$ and for every vector $u \in \mathbb{R}^d$ such that $\|u_{\text{supp}(\beta^*)}\|_1 \geq 0.1 \cdot \|u\|_1$ and for every subset $S \subseteq [n]$ with $|S| \geq n - m$, it holds that $\|(Xu)_S\| \geq \frac{1}{2}\|Xu\|$.

Denote $F_2(\beta) := \sum_{i=1}^n f_2\big(y_i - \langle X_i, \beta \rangle\big)$, where $X_i$ are the rows of $X$, and $f_2$ is as in Eq. (2.1). We devise our estimator for sparse regression and state its statistical guarantees below:

$$\hat{\beta} := \arg\min_{\beta \in \mathbb{R}^d}\Big(F_2(\beta) + 100\sqrt{n \log d} \cdot \|\beta\|_1\Big). \tag{2.3}$$

**Theorem 2.2.** *Let $\beta^* \in \mathbb{R}^d$ be an unknown $k$-sparse vector and let $X \in \mathbb{R}^{n \times d}$ be a deterministic matrix such that for each column $X^i$ of $X$, $\|X^i\| \leq \sqrt{n}$, satisfying the RE-property with $\lambda > 0$ and well-spreadness property with $m \gtrsim \frac{k \log d}{\lambda \cdot \alpha^2}$ (recall that $n \geq m$).*

*Further, let $\eta$ be an $n$-dimensional random vector with independent, symmetrically distributed (about zero) entries and $\alpha = \min_{i \in [n]} \mathbb{P}\big\{|\eta_i| \leq 1\big\}$.*

*Then with probability at least $1 - d^{-10}$ over $\eta$, given $X$ and $y = X\beta^* + \eta$, the estimator Eq. (2.3) satisfies*

$$\frac{1}{n}\Big\|X\big(\hat{\beta} - \beta^*\big)\Big\|^2 \leq O\left(\frac{1}{\lambda} \cdot \frac{k \log d}{\alpha^2 \cdot n}\right) \quad and \quad \|\hat{\beta} - \beta^*\|^2 \leq O\left(\frac{1}{\lambda^2} \cdot \frac{k \log d}{\alpha^2 \cdot n}\right).$$

There are important considerations when interpreting this theorem. The first is the special case $\eta \sim N(0, \sigma^2)^n$, which satisfies our model for $\alpha = \Theta(1/\sigma)$. For this case, it is well known (e.g.,

---

[7]For a vector $v \in \mathbb{R}^d$ and a set $S \subseteq [d]$, we denote by $v_S$ the restriction of $v$ to the coordinates in $S$.

see (Wai19), section 7.3) that under the same RE assumption, the LASSO estimator achieves a prediction error rate of $O(\frac{\sigma^2}{\lambda} \cdot \frac{k \log d}{n}) = O(\frac{k \log d}{\lambda \cdot \alpha^2 \cdot n})$, matching our result. Moreover, this error rate is essentially optimal. Under a standard assumption in complexity theory ($\mathbf{NP} \not\subset \mathbf{P/poly}$), the RE assumption is necessary when considering polynomial-time estimators (ZWJ14). Further, this also shows that the dependence on the RE constant seems unavoidable. Under mild conditions on the design matrix, trivially satisfied if the rows are i.i.d. Gaussian with covariance $\Sigma$ whose condition number is constant, our guarantees are optimal up to constant factors for *all* estimators if $k \leqslant d^{1-\Omega(1)}$ (e.g. $k \leqslant d^{0.99}$), see (RWY11). This optimality also shows that our bound on the number of samples is best possible since otherwise we would not be able to achieve vanishing error. The (non-sparse version of) well-spreadness property was first used in the context of regression in (dNS21). In the same work the authors also showed that, under oblivious noise assumptions, some weak form of spreadness property is indeed necessary.

The second consideration is the *optimal* dependence on $\alpha$: Theorem 2.2 achieves consistency as long as the fraction of inliers satisfies $\alpha = \omega(\sqrt{k \log d / n})$. To get an intuition, observe that lower bounds for standard sparse regression show that, already for $\eta \sim N(0, \sigma \cdot \mathrm{Id}_n)$, it is possible to achieve consistency only for $n = \omega(\sigma^2 k \log d)$ (if $k \leqslant d^{1-\Omega(1)}$). As for this $\eta$, the number of entries of magnitude at most 1 is $O(n/\sigma)$ with high probability, it follows that for $\alpha = \Theta(1/\sigma) \leqslant O\left(\sqrt{(k \log d)/n}\right)$, no estimator is consistent.

To the best of our knowledge, Theorem 2.2 is the first result to achieve consistency under such minimalistic noise settings and deterministic designs. Previous results (BJKK17; SBRJ19; dNS21) focused on simpler settings of Gaussian design $X$ and deterministic noise, and provide no guarantees for more general models. The techniques for Theorem 2.2 also extend to this case.

**Theorem 2.3.** *Let $\beta^* \in \mathbb{R}^d$ be an unknown $k$-sparse vector and let $X$ be a $n$-by-$d$ random matrix with i.i.d. rows $X_1, \ldots X_n \sim N(0, \Sigma)$ for a positive definite matrix $\Sigma$. Further, let $\eta \in \mathbb{R}^n$ be a deterministic vector with $\alpha \cdot n$ coordinates bounded by 1 in absolute value. Suppose that $n \gtrsim \frac{\nu(\Sigma) \cdot k \log d}{\sigma_{\min}(\Sigma) \cdot \alpha^2}$, where $\nu(\Sigma)$ is the maximum diagonal entry of $\Sigma$ and $\sigma_{\min}(\Sigma)$ is its smallest eigenvalue.*

*Then, with probability at least $1 - d^{-10}$ over $X$, given $X$ and $y = X\beta^* + \eta$, the estimator Eq. (2.3) satisfies*

$$\frac{1}{n}\left\|X\left(\hat{\beta} - \beta^*\right)\right\|^2 \leqslant O\left(\frac{\nu(\Sigma) \cdot k \log d}{\sigma_{\min}(\Sigma) \cdot \alpha^2 \cdot n}\right) \qquad and \qquad \left\|\hat{\beta} - \beta^*\right\|^2 \leqslant O\left(\frac{\nu(\Sigma) \cdot k \log d}{\sigma_{\min}^2(\Sigma) \cdot \alpha^2 \cdot n}\right).$$

Even for standard Gaussian design $X \sim N(0,1)^{n \times d}$, the above theorem improves over previous results, which required sub-optimal sample complexity $n \gtrsim (k/\alpha^2) \cdot \log d \cdot \log\|\beta^*\|$. For non-spherical Gaussian designs, the improvement over state of the art (SBRJ19) is more serious: their algorithm requires $\alpha \geqslant \Omega(1/\log \log n)$, while our Theorem 2.3 doesn't have such restrictions and works for all $\alpha \gtrsim \sqrt{\frac{\nu(\Sigma)}{\sigma_{\min}(\Sigma)} \cdot \frac{k \log d}{n}}$; in many interesting regimes $\alpha$ is allowed to be smaller than $n^{-\Omega(1)}$. The dependence on $\alpha$ is nearly optimal: the estimator is consistent as long as $\alpha \geqslant \omega\left(\sqrt{\frac{\nu(\Sigma)}{\sigma_{\min}(\Sigma)} \cdot \frac{k \log d}{n}}\right)$, and from the discussion after Theorem 2.3, if $\alpha \leqslant O(\sqrt{(k \log d)/n})$, no estimator is consistent.

Note that while we can deal with general covariance matrices, to compare Theorem 2.3 with Theorem 2.2 it is easier to consider $\Sigma$ in a normalized form, when $\nu(\Sigma) \leqslant 1$. This can be easily achieved by scaling $X$. Also note that Theorem 2.2 can be generalized to the case $\|X^i\| \leqslant \sqrt{\nu n}$ for arbitrary $\nu > 0$, and then the error bounds and the bound on $m$ should be multiplied by $\nu$.

The RE-property of Theorem 2.2 is a standard assumption in sparse regression and is satisfied by a large family of matrices. For example, with high probability a random matrix $X$ with i.i.d. rows sampled from $\mathcal{N}(0, \Sigma)$, with positive definite $\Sigma \in \mathbb{R}^{d \times d}$ whose diagonal entries are bounded by 1, satisfies the RE-property with parameter $\Omega(\sigma_{\min}(\Sigma))$ for all subsets of $[d]$ of size $k$ (so for every possible support of $\beta^*$) as long as long as $n \gtrsim \frac{1}{\sigma_{\min}(\Sigma)} \cdot k \log d$ (see (Wai19), section 7.3.3). The well-spread assumption is satisfied for such $X$ with for all sets $S \subset [n]$ of size $m \leqslant cn$ (for sufficiently small $c$) and for all subsets of $[d]$ of size $k$ as long as $n \gtrsim \frac{1}{\sigma_{\min}(\Sigma)} \cdot k \log d$.

## 2.3 Optimal fraction of inliers for principal component analysis under oblivious noise

We show here that the dependence on $\alpha$ we obtain in Theorem 2.1 is information theroetically optimal up to constant factors. Concretely, let $L^*, N, Y, \alpha, \rho$ and $\zeta$ be as in Theorem 2.1, and let $0 < \varepsilon < 1$ and $0 < \delta < 1$. A successful $(\varepsilon, \delta)$-weak recovery algorithm for PCA is an algorithm that takes $Y$ as input and returns a matrix $\hat{L}$ such that $\left\|\hat{L} - L^*\right\|_F \leqslant \varepsilon \cdot \rho$ with probability at least $1 - \delta$.

It can be easily seen that the Huber-loss estimator of Theorem 2.1 fails to be a successful weak-recovery algorithm if $\alpha = o(\sqrt{r/n})$ (for both cases $\zeta \leqslant \rho/n$ and $\rho/n \leqslant \zeta$, we need $\alpha = \Omega(\sqrt{r/n})$.) A natural question to ask is whether the condition $\alpha = \Omega(\sqrt{r/n})$ is necessary in general. The following theorem shows that if $\alpha = o(\sqrt{r/n})$, then weak-recovery is information-theoretically impossible. This means that the (polynomially small) fraction of inliers that the Huber-loss estimator of Theorem 2.1 can deal with is optimal up to a constant factor.

**Theorem 2.4.** *There exists a universal constant $C_0 > 0$ such that for every $0 < \varepsilon < 1$ and $0 < \delta < 1$, if $\alpha := \min_{i,j \in [n]} \mathbb{P}[|N_{i,j}| \leqslant \zeta]$ satisfies $\alpha < C_0 \cdot (1 - \varepsilon^2)^2 \cdot (1 - \delta) \cdot \sqrt{r/n}$, and $n$ is large enough, then it is information-theoretically impossible to have a successful $(\varepsilon, \delta)$-weak recovery algorithm.*

*The problem remains information-theoretically impossible (for the same regime of parameters) even if we assume that $L^*$ is incoherent; more precisely, even if we know that $L^*$ has incoherence parameters that are as good as those of a random flat matrix of rank $r$, the theorem still holds.*

# 3 Techniques

To illustrate our techniques in proving statistical guarantees for the Huber-loss estimator, we first use sparse linear regression as a running example. Then, we discuss how the same ideas apply to principal component analysis. Finally, we also remark our techniques for lower bounds.

## 3.1 Sparse linear regression under oblivious noise

We consider the model of Theorem 2.2. Our starting point to attain the guarantees for our estimator Eq. (2.3), i.e., $\hat{\beta} := \arg\min_{\beta \in \mathbb{R}^d} F_2(\beta) + 100\sqrt{n \log d}\left\|\beta\right\|_1$, is a classical approach for $M$-estimators (see e.g. (Wai19), chapter 9). For simplicity, we will refer to $F_2(\beta)$ as the loss function and to $\left\|\beta\right\|_1$ as the regularizer. At a high level, it consists of the following two ingredients:

(I) an upper bound on some norm of the gradient of the loss function at the parameter $\beta^*$,

(II) a lower bound on the curvature of the loss function (in form of a local strong convexity bound) within a structured neighborhood of $\beta^*$. The structure of this neighborhood can roughly be controlled by choosing the appropriate regularizer.

The key aspect of this strategy is that the strength of the statistical guarantees of the estimator crucially depends on the *directions* and the *radius* in which we can establish lower bounds on the curvature of the function. Since these features are inherently dependent on the landscape of the loss function and the regularizer, they may differ significantly from problem to problem. This strategy has been applied successfully for many related problems such as compressed sensing or matrix completion albeit with standard noise assumptions.[8] Under oblivious noise, (dNS21) used a particular instantiation of this framework to prove optimal convergence of the Huber-loss – without any regularizer – for standard linear regression. Such estimator, however, doesn't impose any structure on the neighborhood of $\beta^*$ considered in (II) and thus, can only be used to obtain sub-optimal guarantees for sparse regression.

In the context of sparse regression, the above two conditions translate to: (I) an upper bound on the largest entry in absolute value of the gradient of the loss function at $\beta^*$, and (II) a lower bound on the curvature of $F_2$ within the set of *approximately $k$-sparse vectors*[9] close to $\beta^*$. We use this recipe to show that *all* approximate minimizers of $F$ are close to $\beta^*$. While the idea of restricting to only approximately sparse directions has also been applied for the LASSO estimator in sparse

---

[8]The term "standard noise assumptions" is deliberately vague; as a concrete example, we will refer to (sub)-Gaussian noise distributions. See again chapter 9 of (Wai19) for a survey.

[9]We will clarify this notion in the subsequent paragraphs.

regression under standard (sub)-Gaussian noise, in the presence of oblivious noise, our analysis of the Huber-loss function requires a more careful approach.

More precisely, under the assumptions of Theorem 2.2, the error bound can be computed as

$$O\left(\frac{s\|G\|_{\text{reg}}^*}{\kappa}\right), \tag{3.1}$$

where $G$ is a gradient of Huber loss at $\beta^*$, $\|\cdot\|_{\text{reg}}^*$ is a norm dual to the regularization norm (which is equal to $\|\cdot\|_{\max}$ for $\ell_1$ regularizer), $s$ is a *structure* parameter, which is equal to $\sqrt{k/\lambda}$, and $\kappa$ is a restricted strong convexity parameter. Note that by error here we mean $\frac{1}{\sqrt{n}}\|X(\hat{\beta} - \beta^*)\|$.

Similarly, under under assumptions of Theorem 2.1, we get the error bound Eq. (3.1), where $G$ is a gradient of Huber loss at $L^*$, $\|\cdot\|_{\text{reg}}^*$ is a norm dual to the nuclear norm (i.e. the spectral norm), structure parameter $s$ is $\sqrt{r}$, and $\kappa$ is a restricted strong convexity parameter. For more details on the conditions of the error bound, see the supplementary material. Below, we explore the bounds on the norm of the gradient and on the restricted strong convexity parameter.

**Bounding the gradient of the Huber loss.** The gradient of the Huber-loss $F_2(\cdot)$ at $\beta^*$ has the form $\nabla F_2(\beta^*) = \sum_{i=1}^n f_2'[\eta_i] \cdot X_i$, where $X_i$ is the $i$-th row of $X$. The random variables $f_2'[\eta_i]$, $i \in [n]$, are independent, centered, symmetric and bounded by 2. Since we assume that each column of $X$ has norm at most $\sqrt{n}$, the entries of the row $X_i$ are easily bounded by $\sqrt{n}$. Thus, $\nabla F_2(\beta^*)$ is a vector with independent, symmetric entries with bounded variance, so its behavior can be easily studied through standard concentration bounds. In particular, by a simple application of Hoeffding's inequality, we obtain, with high probability,

$$\left\|\nabla F_2(\beta^*)\right\|_{\max} = \max_{j \in [d]} \left| \sum_{i \in [n]} f_2'[\eta_i] \cdot X_{ij} \right| \leq O\left(\sqrt{n \log d}\right). \tag{3.2}$$

**Local strong convexity of the Huber loss.** Proving local strong convexity presents additional challenges. Without the sparsity constraint, (dNS21) showed that under a slightly stronger spreadness assumptions than Theorem 2.2, the Huber loss is locally strongly convex within a constant radius $R$ ball centered at $\beta^*$ whenever $n \gtrsim d/\alpha^2$. (This function is not globally strongly convex due to its linear parts.) Using their result as a black-box, one can obtain the error guarantees of Theorem 2.2, but with suboptimal sample complexity. The issue is that with the substantially smaller sample size of $n \geq \tilde{O}(k/\alpha^2)$ that resembles the usual considerations in the context of sparse regression, the Huber loss is *not* locally strongly convex around $\beta^*$ uniformly across *all directions*, so we cannot hope to prove convergence with optimal sample complexity using this argument. To overcome this obstacle, we make use of the framework of M-estimators: Since we consider a regularized version of the Huber loss, it will be enough to show local strong convexity in a radius $R$ uniformly across all directions which are *approximately $k$-sparse*. For this substantially weaker condition, $\tilde{O}(k/\alpha^2)$ will be enough.

More in details, for observations $y = X\beta^* + \eta$ and an arbitrary $u \in \mathbb{R}^d$ of norm $\|u\| \leq R$, it is possible to lower bound the Hessian[10] of the Huber loss at $\beta^* + u$ by:[11]

$$HF_2(\beta^* + u) = \sum_{i=1}^n f_2''[(Xu)_i - \eta_i] \cdot X_i X_i^{\mathsf{T}} = \sum_{i=1}^n \mathbf{1}_{[|(Xu)_i - \eta_i| \leq 2]} \cdot X_i X_i^{\mathsf{T}}$$
$$\geq M(u) := \sum_{i=1}^n \mathbf{1}_{[|\langle X_i, u \rangle| \leq 1]} \cdot \mathbf{1}_{[|\eta_i| \leq 1]} \cdot X_i X_i^{\mathsf{T}}$$

As can be observed, we do not attempt to exploit cancellations between $Xu$ and $\eta$. Let $Q := \{i \in [n] \mid |\eta_i| \leq 1\}$ be the set of uncorrupted entries of $\eta$. Given that with high probability $Q$ has size $\Omega(\alpha \cdot n)$, the best outcome we can hope for is to provide a lower bound of the form $\langle u, M(u)u \rangle \geq \alpha n$ in the direction $\hat{\beta} - \beta^*$. In (dNS21), it was shown that if the span of the measurement matrix $X$ is well spread, then $\langle u, M(u)u \rangle \geq \Omega(\alpha \cdot n)$.

---

[10]The Hessian does not exist everywhere. Nevertheless, the second derivative of the penalty function $f_2$ exists as an $L_1$ function in the sense that $f_2'(b) - f_2'(a) = 2 \int_a^b \mathbf{1}_{[|t| \leq 2]} dt$. This property is enough for our purposes.

[11]A more extensive explanation of the first part of this analysis can be found in (dNS21).

If the direction $\hat{\beta} - \beta^*$ was fixed, it would suffice to show the curvature in that single direction through the above reasoning. However, $\hat{\beta}$ depends on the unknown random noise vector $\eta$. Without the regularizer in Eq. (2.3), this dependence indicates that the vector $\hat{\beta}$ may take any possible direction, so one needs to ensure local strong convexity to hold in a constant-radius ball centered at $\beta^*$. That is, $\min_{\|u\| \leqslant R} \lambda_{\min}(M(u)) \geqslant \Omega(\alpha \cdot n)$. It can be shown through a covering argument (of the ball) that this bound holds true for $n \geqslant d/\alpha^2$. This is the approach of (dNS21).

**The minimizer of the Huber loss follows a sparse direction.** The main issue with the above approach is that no information concerning the direction $\hat{\beta} - \beta^*$ is used. In the settings of sparse regression, however, our estimator contains the regularizer $\|\beta\|_1$. The main consequence of the regularizer is that the *direction* $\hat{\beta} - \beta^*$ is approximately flat in the sense $\|\hat{\beta} - \beta^*\|_1 \leqslant O\left(\sqrt{k}\|\hat{\beta} - \beta^*\|\right)$. The reason[12] is that due to the structure of the objective function in Eq. (2.3) and concentration of the gradient Eq. (3.2), the penalty for dense vectors is larger than the absolute value of the inner product $\langle \nabla F(\beta^*), \hat{\beta} - \beta^* \rangle$ (which, as previously argued, concentrates around its (zero) expectation).

This specific structure of the minimizer implies that it suffices to prove local strong convexity *only* in approximately sparse directions. For these set of directions, we carefully construct a sufficiently small covering set so that $n \geqslant \tilde{O}(k/\alpha^2)$ samples suffice to ensure local strong convexity over it.

*Remark* 3.1 (Comparison with LASSO). it is important to remark that while this approach of only considering approximately sparse directions has also been used in the context of sparse regression under Gaussian noise (e.g. the LASSO estimator), obtaining the desired lower bound is considerably easier in these settings as it directly follows from the restricted eigenvalue property of the design matrix. In our case, we require an additional careful probabilistic analysis which uses a covering argument for the set of approximately sparse vectors. As we see however, it turns out that we do not need any additional assumptions on the design matrix when compared with the LASSO estimator except for the well-spreadness property (recall that some weak version of well-spreadness is indeed necessary in robust settings, see (dNS21)).

## 3.2 Principal component analysis under oblivious noise

A convenient feature of the approach in Section 3.1 for sparse regression, is that it can be easily applied to additional problems. We briefly explain here how to apply it for principal component analysis. We consider the model defined in Theorem 2.1. We use an estimator based on the Huber loss equipped with the nuclear norm as a regularizer to enforce the low-rank structure in our estimator

$$\hat{L} := \underset{L \in \mathbb{R}^{n \times n}, \, \|L\|_{\max} \leqslant \rho/n}{\operatorname{argmin}} \left( F_{\zeta + \rho/n}(Y - L) + 100\sqrt{n}\left(\zeta + \rho/n\right)\|L\|_{\text{nuc}} \right). \tag{3.3}$$

In this setting, the gradient $\nabla F_{\zeta + \rho/n}(Y - L^*)$ is a matrix with independent, symmetric entries which are bounded (by $\zeta + \rho/n$) and hence its spectral norm is $O\left((\zeta + \rho/n)\sqrt{n}\right)$ with high probability. Local strong convexity can be obtained in a similar fashion as shown in Section 3.1: due to the choice of the Huber transition point all entries with small noise are in the quadratic part of $F$. Moreover, the nuclear norm regularizer ensures that the minimizer is an approximately low-rank matrix in the sense that $\|M\|_{\text{nuc}} \leqslant O\left(\sqrt{r}\|M\|_{\text{F}}\right)$. So again, it suffices to provide curvature of the loss function only on these subset of structured directions.

*Remark* 3.2 (Incoherence vs. spikiness). Recall the discussion on incoherence in Section 2. If for every $\mu \leqslant n/\log^2 n$ matrix $L^*$ doesn't satisfy $\mu$-incoherence conditions, the results in (CLMW11; ZLW+10) cannot be applied. However, our estimator achieves error $\|\hat{L} - L^*\|_{\text{F}}/\|L^*\|_{\text{F}} \to 0$ as $n \to \infty$. Indeed, let $\omega(1) \leqslant f(n) \leqslant o(\log^2 n)$ and assume $\zeta = 0$. Let $u \in \mathbb{R}^n$ be an $f(n)$-sparse unit vector whose nonzero entries are equal to $1/\sqrt{f(n)}$. Let $v \in \mathbb{R}^n$ be a vector with all entries equal to $1/\sqrt{n}$. Then, $uv^\mathsf{T}$ does not satisfy incoherence with any $\mu < n/f(n)$. We have $\|uv^\mathsf{T}\|_{\text{F}} = 1$, and the error of our estimator is $O(1/(\alpha\sqrt{f(n)}))$, so it tends to zero for constant (or even some subconstant) $\alpha$.

Furthermore notice that the dependence of the error of Theorem 2.1 on the maximal entry of $L^*$ is inherent if we do not require incoherence. Indeed, consider $L_1 = b \cdot e_1 e_1^\mathsf{T}$ for large enough $b > 0$

---

[12]This phenomenon is a consequence of the *decomposability* of the $L_1$ norm, see the supplementary material.

and $L_2 = e_2 e_2^\mathsf{T}$. For constant $\alpha$, let $|N_{ij}|$ be 1 with probability $\alpha/2$, 0 with probability $\alpha/2$ and $b$ with probability $1 - \alpha$. Then given $Y$ we cannot even distinguish between cases $L^* = L_1$ or $L^* = L_2$, and since $\|L_1 - L_2\|_\mathrm{F} \geqslant b$, the error should also depend on $b$.

*Remark* 3.3 ($\alpha$ vs $\alpha^2$: what if one knows which entries are corrupted?). As was observed in Section 2, the error bound of our estimator is worse than the error for matrix completion by a factor $1/\sqrt{\alpha}$. We observe similar effect in linear regression: if, as in matrix completion, we are given a randomly chosen $\alpha$ fraction of observations $\left\{ \left( X_i, y_i = \langle X_i, \beta^* \rangle + \eta_i \right) \right\}_{i=1}^n$ where $\eta \sim N(0,1)^n$, and since for the remaining samples we may not assume any bound on the signal-to-noise ratio, then this problem is essentialy the same as linear regression with $\alpha n$ observations. Thus the optimal prediction error rate is $\Theta(\sqrt{d/(\alpha n)})$. Now, if we have $y = X\beta^* + \eta$, where $\eta \sim N(0, 1/\alpha^2)$, then with probability $\Theta(\alpha)$, $|\eta_i| \leqslant 1$, but the optimal prediction error rate in this case is $\Theta(\sqrt{d/(\alpha^2 n)})$. So in both linear regression and robust PCA, *prior knowledge* of the set of corrupted entries makes the problem easier.

## 3.3 Optimal fraction of inliers for principal component analysis under oblivious noise

In order to prove Theorem 2.4, we will adopt a generative model for the hidden matrix $L^*$: We will generate $L^*$ randomly but assume that the distribution is known to the algorithm. This makes the problem easier. Therefore, any impossibility result for this generative model would imply impossibility for the more restrictive model for which $L^*$ is deterministic but unknown.

We generate a random flat matrix $L^*$ using $n \cdot r$ independent and uniform random bits in such a way that $L^*$ is of rank $r$ and incoherent with high probability. Then, for every constant $0 < \xi < 1$, we find a distribution for the random noise $N$ in such a way that the fraction of inliers satisfies $\alpha := \mathbb{P}[|N_{ij}| \leqslant \zeta] = \Theta\left( \xi \sqrt{r/n} \right)$, and such that the mutual information between $L^*$ and $Y = L^* + N$ can be upper bounded as $I(L^*; Y) \leqslant O(\xi \cdot n \cdot r)$. Roughly speaking the smaller $\xi$ gets, the more independent $L^*$ and $Y$ will be. Now using an inequality that is similar to the standard Fano-inequality but adapted to weak-recovery, we show that if there is a successful $(\varepsilon, \delta)$-weak recovery algorithm for $L^*$ and $N$, then $I(L^*; Y) \geqslant \Omega\left( (1 - \varepsilon^2)^2 \cdot (1 - \delta) \cdot n \cdot r \right)$. By combining all these observations together, we can deduce that if $\xi$ is small enough, it is impossible to have a successful $(\varepsilon, \delta)$-weak recovery algorithm for $L^*$ and $N$.

## Acknowledgments and Disclosure of Funding

The authors thank the anonymous reviewers for useful comments. This project has received funding from the European Research Council (ERC) under the European Union's Horizon 2020 research and innovation programme (grant agreement No 815464).

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
