## Outline of Appendices

In Appendix A, we introduce some basic notation and definitions used throughout all of the appendices. In Appendix B, we introduce a general framework that allows us to prove our main theorems about consistent estimation (Theorem 2.1, Theorem 2.2 and Theorem 2.3).

Appendices C to F are devoted to the proofs of theorems from Section 2. Concretely, in Appendix C we prove Theorem 2.1, Appendix D contains a proof of Theorem 2.2 and in Appendix E we prove Theorem 2.3. Complementing these algorithmic results, we prove a lower bound (Theorem 2.4) in Appendix F.

Finally, Appendix G contains the proofs of a few facts about the Huber loss and Appendix H contains a few facts from probability theory.

## A    Preliminaries

**Notation.**    For $\beta \in \mathbb{R}^d$ we define the function $\|\beta\|_0 := \sum_{i \in [d]} \mathbf{1}_{[\beta_i \neq 0]}$ . For a subspace $\Omega \subseteq \mathbb{R}^d$ we denote the projection of $\beta$ onto $\Omega$ by $\beta_\Omega$. We write $\Omega^\perp$ for the orthogonal complement of $\Omega$. For $N \in \mathbb{N}$ we denote $[N] := \{1, 2, \dots, N\}$. We write log for the logarithm to the base $e$.

For a matrix $X \in \mathbb{R}^{d \times d}$ we denote by $\mathrm{rspan}(X)$ and $\mathrm{cspan}(X)$ respectively the rows and columns span of $X$, and we write $\|X\|$ for the spectral norm of $X$, $\|X\|_F$ for its Frobenius norm, $\|X\|_{\mathrm{nuc}}$ for its nuclear norm and $\|X\|_{\max} := \max_{i,j \in [n]} |X_{ij}|$. For a vector $v \in \mathbb{R}^N$ we write $\|v\|$ for its Euclidean norm, $\|v\|_1 = \sum_{i=1}^N |v_i|$ and $\|v_i\|_\infty = \max_{i \in [N]} |v_i|$. For a norm $\|\cdot\|$ we write $\|\cdot\|^*$ for its dual. We denote by $G \sim N(0, 1)^{n \times d}$ a random $n$-by-$d$ matrix $G$ with i.i.d. standard Gaussian entries. Similarly, we denote by $g \sim N(0, 1)^n$ an $n$-dimensional random vector $g$ with i.i.d. standard Gaussian entries.

For a set $\mathcal{S}$ and a metric $\rho : \mathcal{S} \times \mathcal{S} \to [0, \infty)$, we denote an $\varepsilon$-net in $\mathcal{S}$ by $\mathcal{N}_{\varepsilon, \rho}(\mathcal{S})$. That is, $\mathcal{N}_{\varepsilon, \rho}(\mathcal{S})$ is a subset of $\mathcal{S}$ such that for any $u \in \mathcal{S}$ there exists $v \in \mathcal{N}_{\varepsilon, \rho}(\mathcal{S})$ satisfying $\rho(u, v) \leqslant \varepsilon$.

## B    Meta-Theorem

We present a high-level theorem which will be applied to prove Theorem 2.1, Theorem 2.2 and Theorem 2.3. Recall the general setting an estimation problem: we start with a family of probability distributions $\mathcal{P} := \{\mathbb{P}_\theta \mid \theta \in \Omega\}$ over some space $\mathcal{Z}$ and indexed by some parameter $\theta \in \Omega$. We observe a collection of $n$ *independent* samples $Z = (Z_1, \dots, Z_n)$ taking value in $\mathcal{Z}$, drawn from an unknown probability distribution $\mathbb{P}_{\theta^*} \in \mathcal{P}$. We assume $\Omega \subseteq \mathbb{R}^d$ and $\mathcal{Z} \subseteq \mathbb{R}^D$ for some integers $d$ and $D$. Our goal is then to recover $\theta^*$. That is, given $Z$, the goal is to find $\hat{\theta} \in \mathbb{R}^d$ such that for some suitable error function $\mathcal{E} : \mathbb{R}^d \to [0, \infty)$, the value $\mathcal{E}(\theta^* - \hat{\theta})$ is as small as possible. It is clear that this general setting also captures settings in which the observations are perturbed by oblivious adversarial noise.

On a high level, we will use the following scheme:

1. Let $\|\cdot\|_{\mathrm{reg}} : \mathbb{R}^d \to [0, \infty)$ be a norm, and let $\gamma \in \mathbb{R}$ be a scalar. Design a cost function $F : \mathbb{R}^d \to \mathbb{R}_{\geqslant 0}$ which depends on $Z$ .

2. For a set $C \subseteq \mathbb{R}^d$ , show that the target parameter (or some approximation of it)

$$\hat{\theta} := \arg\min_{\theta \in C} \big( F(\theta) + \gamma \|\theta\|_{\mathrm{reg}} \big)$$

   satisfies $\mathcal{E}(\theta^* - \hat{\theta}) \leqslant R$ for some acceptable $R \geqslant 0$ with high probability over the samples $Z$ .

3. Argue that $\hat{\theta}$ can be computed efficiently.

The norm $\|\cdot\|_{\mathrm{reg}}$ is often referred to as a *regularizer*. Its role is to enforce a certain structure on the target parameter. For example, in the context of sparse linear regression $y = X\beta^* + \eta$ with $\beta^* \in \mathbb{R}^d$ being a $k$-sparse vector, the LASSO estimator: $\hat{\beta} := \arg\min_{\beta \in \mathbb{R}^d} \big( \|X\beta - y\|^2 + \gamma \|\beta\|_1 \big)$ follows the

description above. In this example, the cost function is the squared euclidean norm and the regularizer corresponds to a convex relaxation of the norm $\|\beta\|_0$.

If the cost function and the set $C$ are convex and satisfy mild assumptions, the estimator can be computed efficiently (in polynomial time). The estimators that we use for PCA and sparse linear regression can be computed in polynomial time. For more details on computational aspects of convex optimization, see (Vis18).

For convex cost functions the meta-theorem below (which appears in different forms in the literature, e.g. see (Wai19), section 9.4) can be used to mechanically bound the guarantees of the estimator. Before stating the theorem, let's define the following set: for a norm $\|\cdot\|_{\text{reg}}$ and for a vector subspace $V \subseteq \mathbb{R}^d$ and $b \geq 1$, we denote

$$\mathcal{S}_b(V) = \left\{ u \in \mathbb{R}^d \mid \|u\|_{\text{reg}} \leq b\|u_V\|_{\text{reg}} \right\},$$

where $u_V$ is the orthogonal projection of $u$ on $V$.

**Theorem B.1.** *Let $\gamma, \kappa, R, s$ be positive real numbers and let $C \subseteq \mathbb{R}^d$ be a convex set. Consider a vectors space $\Omega \subseteq \mathbb{R}^d$ and let $\theta^* \in \Omega \cap C$.*

*Let $\|\cdot\|_{\text{reg}} : \mathbb{R}^d \to [0, \infty)$ be a norm and consider a continuous error function $\mathcal{E} : \mathbb{R}^d \to [0, \infty)$ such that $\mathcal{E}(0) = 0$. Let $F : \mathbb{R}^d \to \mathbb{R}$ be a convex differentiable cost function.*

*Suppose that there exists a vector space $\overline{\Omega}$ such that $\Omega \subseteq \overline{\Omega} \subseteq \mathbb{R}^d$ and such that the following properties hold:*

*(Decomposability) For all $u \in \Omega$ and $v \in \overline{\Omega}^\perp$,*

$$\|v + u\|_{\text{reg}} = \|v\|_{\text{reg}} + \|u\|_{\text{reg}}. \tag{B.1}$$

*(Contraction) For all $u \in \mathcal{S}_4(\overline{\Omega})$,*

$$\|u\|_{\text{reg}} \leq s \cdot \mathcal{E}(u). \tag{B.2}$$

*(Gradient bound) The dual norm of $\|\cdot\|_{\text{reg}}$ of gradient of $F$ at $\theta^*$ satisfies*

$$\|\nabla F(\theta^*)\|_{\text{reg}}^* \leq \gamma/2. \tag{B.3}$$

*(Restricted local strong convexity) Let $\mathcal{B}_R := \left\{ u \in \mathbb{R}^d \mid \mathcal{E}(u) = R, \theta^* + u \in C \right\}$. Then*

$$\forall u \in \mathcal{B}_R \cap \mathcal{S}_4(\overline{\Omega}) \qquad F(\theta^* + u) \geq F(\theta^*) + \langle \nabla F(\theta^*), u \rangle + \frac{\kappa}{2}(\mathcal{E}(u))^2. \tag{B.4}$$

*(Bound on radius) Parameters $\gamma, \kappa, R$ and $s$ satisfy*

$$\frac{\gamma \cdot s}{\kappa} \leq R/4. \tag{B.5}$$

*Then, for every $\theta' \in C$ such that $F(\theta') + \gamma\|\theta'\|_{\text{reg}} \leq F(\theta^*) + \gamma\|\theta^*\|_{\text{reg}}$,*

$$\mathcal{E}(\theta' - \theta^*) < R.$$

For completeness, we include a proof of Theorem B.1. We will need the following lemma.

**Lemma B.2.** *Consider the settings of Theorem B.1. If $\theta \in C$ satisfies*

$$F(\theta) + \gamma\|\theta\|_{\text{reg}} \leq F(\theta^*) + \gamma\|\theta^*\|_{\text{reg}},$$

*then $\theta - \theta^* \in \mathcal{S}_4(\overline{\Omega})$.*

*Proof.* Denote $\Delta = \theta - \theta^*$. By the decomposability of the regularizer Eq. (B.1),

$$\|\theta^* + \Delta\|_{\text{reg}} = \left\|\theta_\Omega^* + \Delta_{\overline{\Omega}} + \Delta_{\overline{\Omega}^\perp}\right\|_{\text{reg}}$$

$$\geqslant \left\|\theta_\Omega^* + \Delta_{\overline{\Omega}^\perp}\right\|_{\text{reg}} - \left\|\Delta_{\overline{\Omega}}\right\|_{\text{reg}} \qquad \text{(Triangle Inequality)}$$

$$= \left\|\theta_\Omega^*\right\|_{\text{reg}} + \left\|\Delta_{\overline{\Omega}^\perp}\right\|_{\text{reg}} - \left\|\Delta_{\overline{\Omega}}\right\|_{\text{reg}}. \qquad \text{(Decomposability of } \|\cdot\|_{\text{reg}} \text{)}.$$

By convexity of the cost function and Hölder's inequality,

$$F(\theta^* + \Delta) - F(\theta^*) \geqslant -|\langle \nabla F(\theta^*), \Delta \rangle| \geqslant -\|\nabla F(\theta^*)\|_{\text{reg}}^* \cdot \|\Delta\|_{\text{reg}}.$$

Hence by the gradient bound and the decomposability of the regularizer,

$$F(\theta^* + \Delta) - F(\theta^*) \geqslant -\frac{\gamma}{2} \cdot \|\Delta\|_{\text{reg}} = -\frac{\gamma}{2}\left(\left\|\Delta_{\overline{\Omega}}\right\|_{\text{reg}} + \left\|\Delta_{\overline{\Omega}^\perp}\right\|_{\text{reg}}\right).$$

Recall that $F(\theta^* + \Delta) + \gamma\|\theta^* + \Delta\|_{\text{reg}} \leqslant F(\theta^*) + \gamma\|\theta^*\|_{\text{reg}}$, hence

$$0 \geqslant \gamma\left(\|\theta^* + \Delta\|_{\text{reg}} - \left\|\theta_\Omega^*\right\|_{\text{reg}}\right) + (F(\theta^* + \Delta) - F(\theta^*))$$

$$\geqslant \gamma\left(\|\theta^* + \Delta\|_{\text{reg}} - \left\|\theta_\Omega^*\right\|_{\text{reg}}\right) - \frac{\gamma}{2}\left(\left\|\Delta_{\overline{\Omega}}\right\|_{\text{reg}} + \left\|\Delta_{\overline{\Omega}^\perp}\right\|_{\text{reg}}\right)$$

$$\geqslant \gamma\left(\left\|\Delta_{\overline{\Omega}^\perp}\right\|_{\text{reg}} - \left\|\Delta_{\overline{\Omega}}\right\|_{\text{reg}}\right) - \frac{\gamma}{2}\left(\left\|\Delta_{\overline{\Omega}}\right\|_{\text{reg}} + \left\|\Delta_{\overline{\Omega}^\perp}\right\|_{\text{reg}}\right)$$

$$= \frac{\gamma}{2}\left(\left\|\Delta_{\overline{\Omega}^\perp}\right\|_{\text{reg}} - 3\left\|\Delta_{\overline{\Omega}}\right\|_{\text{reg}}\right).$$

Therefore, we have $\left\|\Delta_{\overline{\Omega}^\perp}\right\|_{\text{reg}} \leqslant 3\left\|\Delta_{\overline{\Omega}}\right\|_{\text{reg}}$, and thus

$$\|\Delta\|_{\text{reg}} \leqslant \left\|\Delta_{\overline{\Omega}^\perp}\right\|_{\text{reg}} + \left\|\Delta_{\overline{\Omega}}\right\|_{\text{reg}} \leqslant 4\left\|\Delta_{\overline{\Omega}}\right\|_{\text{reg}}.$$

$\square$

We are now ready to prove the theorem.

*Proof of Theorem B.1.* Denote $G(\theta) = F(\theta) + \gamma\|\theta\|_{\text{reg}}$.

Assume by contradiction that there exists $\theta' \in C$ such that $\mathcal{E}(\theta' - \theta^*) \geqslant R$ and $G(\theta') \leqslant G(\theta^*)$. By continuity of $\mathcal{E}$, there should exist a point $\tilde{\theta}$ on the segment between $\theta'$ and $\theta^*$ such that $\mathcal{E}(\tilde{\theta} - \theta^*) = R$. Since $C$ is convex, $\tilde{\theta} \in C$, so $\tilde{\theta} - \theta^* \in \mathcal{B}_R$. By convexity of $G$, $G(\tilde{\theta}) \leqslant G(\theta^*)$. Denote $\tilde{\Delta} = \tilde{\theta} - \theta^*$. We get

$$F(\theta^* + \tilde{\Delta}) - F(\theta^*) \leqslant \gamma\left(\|\theta^*\|_{\text{reg}} - \|\tilde{\Delta} + \theta^*\|_{\text{reg}}\right) \qquad \text{(Definition of } \tilde{\Delta} \text{ \& } G(\tilde{\theta}) \leqslant G(\theta^*))$$

$$\leqslant \gamma \cdot \|\tilde{\Delta}\|_{\text{reg}} \qquad \text{(Triangle Inequality)}$$

$$\leqslant \gamma \cdot s \cdot \mathcal{E}(\tilde{\Delta}). \qquad \text{(Lemma B.2 \& Contraction (Eq. (B.2)))}$$

By restricted local strong convexity (Eq. (B.4)) and the Gradient bound (Eq. (B.3)), we have

$$(\mathcal{E}(\tilde{\Delta}))^2 \leqslant \frac{2}{\kappa}\left(|\langle \nabla F(\theta^*), \Delta \rangle| + (F(\theta^* + \tilde{\Delta}) - F(\theta^*))\right) \qquad \text{(Eq. (B.4))}$$

$$\leqslant \frac{2}{\kappa}\left(|\langle \nabla F(\theta^*), \Delta \rangle| + \gamma \cdot s \cdot \mathcal{E}(\tilde{\Delta})\right) \qquad \left(F(\theta^* + \tilde{\Delta}) - F(\theta^*) \leqslant \gamma \cdot s \cdot \mathcal{E}(\tilde{\Delta})\right)$$

$$\leqslant \frac{2}{\kappa}\left(\|\nabla F(\theta^*)\|_{\text{reg}}^* \|\tilde{\Delta}\|_{\text{reg}} + \gamma \cdot s \cdot \mathcal{E}(\tilde{\Delta})\right) \qquad \text{(Hölder's inequality)}$$

$$< 4 \cdot \frac{\gamma \cdot s \cdot \mathcal{E}(\tilde{\Delta})}{\kappa} \qquad \text{(Eq. (B.3) \& Lemma B.2 \& Eq. (B.2))}$$

$$\leqslant R \cdot \mathcal{E}(\tilde{\Delta}). \qquad \text{(Eq. (B.5))}$$

So $\mathcal{E}(\tilde{\Delta}) < R$, leading to a contradiction. Hence every $\theta' \in C$ such that $G(\theta') \leqslant G(\theta^*)$ satisfies $\mathcal{E}(\theta' - \theta^*) < R$.

$\square$

# C Principal component analysis with oblivious outliers (Theorem 2.1)

We will prove Theorem 2.1, that we restate in this section

Recall that for $L \in \mathbb{R}^{n \times n}$, $F_h(L) = \sum_{i,j \in [n]} f_h(L_{ij})$, where

$$f_h(t) := \begin{cases} \frac{1}{2}t^2 & \text{for } |t| \leq h\,, \\ h(|t| - \frac{h}{2}) & \text{otherwise.} \end{cases}$$

**Theorem** (Restatement of Theorem 2.1). *Let $L^* \in \mathbb{R}^{n \times n}$ be an unknown deterministic matrix, let $N^* \in \mathbb{R}^{n \times n}$ be a random matrix with independent, symmetrically distributed (about zero) entries and let $\alpha := \min_{i,j \in [n]} \mathbb{P}\{|N_{ij}| \leq \zeta\}$ for some $\zeta \geq 0$. Suppose that $\mathrm{rank}(L^*) = r$ and $\|L^*\|_{\max} \leq \rho/n$.*

*Consider the following estimator:*

$$\hat{L} := \underset{L \in \mathbb{R}^{n \times n},\ \|L\|_{\max} \leq \rho/n}{\mathrm{argmin}} \left(F_h(Y - L) + \gamma\|L\|_{\mathrm{nuc}}\right), \tag{C.1}$$

*where $h = \zeta + \rho/n$ and $\gamma = 100\sqrt{n}(\zeta + \rho/n)$.*

*Then, with probability at least $1 - 2^{-n}$ over $N$, given $Y = L^* + N$, $\zeta$ and $\rho$, the estimator $\hat{L}$ satisfies*

$$\left\|\hat{L} - L^*\right\|_{\mathrm{F}} \leq O\left(\frac{\sqrt{rn}}{\alpha}\right) \cdot (\zeta + \rho/n)\,.$$

In light of Theorem B.1, we can prove Theorem 2.1 by showing that the estimator $\hat{L}$ in Eq. (C.1) fulfills all the conditions of Theorem B.1 with $F(L) := F_h(Y - L) = F_{\zeta+\rho/n}(Y - L)$, $\|\cdot\|_{\mathrm{reg}} := \|\cdot\|_{\mathrm{nuc}}$, $\gamma = 100\sqrt{n}(\zeta + \rho/n)$ and $\mathcal{E}(\cdot) := \|\cdot\|_{\mathrm{F}}$.

To this end, we define the two vector spaces in Theorem B.1, $\Omega$ and $\overline{\Omega}$, as follows:

$$\Omega := \left\{L \in \mathbb{R}^{n \times n} \mid \mathrm{rspan}(L) \subseteq \mathrm{rspan}(L^*)\,, \mathrm{cspan}(L) \subseteq \mathrm{cspan}(L^*)\right\}, \tag{C.2}$$

$$\overline{\Omega}^{\perp} := \left\{L \in \mathbb{R}^{n \times n} \mid \mathrm{rspan}(L) \subseteq \mathrm{rspan}(L^*)^{\perp}\,, \mathrm{cspan}(L) \subseteq \mathrm{cspan}(L^*)^{\perp}\right\}. \tag{C.3}$$

It is easy to see that $\Omega \subseteq \overline{\Omega}$ and the nuclear norm is *decomposable* per Eq. (B.1) with respect to $\Omega$ and $\overline{\Omega}^{\perp}$. That is, for all $L \in \Omega$ and $L' \in \overline{\Omega}^{\perp}$, we have $\|L + L'\|_{\mathrm{nuc}} = \|L\|_{\mathrm{nuc}} + \|L'\|_{\mathrm{nuc}}$, satisfying condition Eq. (B.1).

Moreover, since $L^*$ has rank $r$, Eq. (C.3) implies that any matrix in $\overline{\Omega}$ has rank at most $2r$. Hence, we immediately obtain that for all $L \in \mathcal{S}_4(\overline{\Omega}) = \left\{L \in \mathbb{R}^{n \times n} \mid \|L\|_{\mathrm{nuc}} \leq 4\|L_{\overline{\Omega}}\|_{\mathrm{nuc}}\right\}$, $\|L\|_{\mathrm{nuc}} \leq 4\sqrt{2r}\|L\|_{\mathrm{F}}$, satisfying condition Eq. (B.2) with $s = 4\sqrt{2r}$.

It remains to prove the gradient bound of the condition Eq. (B.3), i.e., a bound on the spectral norm of $\nabla F_h(Y - L^*)$ (since the dual norm of the nuclear norm is the spectral norm), and the local strong convexity of the condition Eq. (B.4).

We start with proving the gradient bound:

**Lemma C.1** (Gradient bound of spectral norm). *Consider the settings of Theorem 2.1, and let $\delta \in (0,1)$. Then with probability at least $1 - \delta/2$,*

$$\|\nabla F_h(Y - L^*)\| \leq 10h\sqrt{n + \log(2/\delta)}\,.$$

*Proof.* By definition of the Huber penalty for all $i, j \in [n]$

$$-h \leq \nabla f_h(Y_{ij} - L^*_{ij}) = \nabla f_h(N_{ij}) \leq h\,.$$

That is, entries are independent, symmetric and bounded by $h$ in absolute value. Hence by Fact H.7, with probability $1 - \delta/2$ the spectral norm of this matrix is bounded by $10h\sqrt{n + \log(2/\delta)}$. $\qquad\square$

**Proof of local strong convexity.** We first bound the size of an $\varepsilon$-*net* for the set of approximately low-rank matrices (Lemma C.2) and then apply this bound to derive a lower bound for the second-order integral of the Huber-loss function with penalty $h$ (Lemma C.3).

**Lemma C.2** ($\varepsilon$-Net for approximately low-rank matrices). *Let* $0 < \varepsilon < 1$ *and* $s \geqslant 1$. *Define*

$$\mathcal{L}_s := \left\{ L \in \mathbb{R}^{n \times n} \;\middle|\; \|L\|_{\mathrm{nuc}} \leqslant s \|L\|_{\mathrm{F}}, \|L\|_{\mathrm{F}} \leqslant 1 \right\}.$$

*Then* $\mathcal{L}_s$ *has an* $\varepsilon$-*net of size* $\exp\left[ \frac{16s^2 n}{\varepsilon^2} \right]$.

*Proof.* Let $W$ be a $n$-by-$n$ random matrix with i.i.d entries $W_{ij} \sim N(0, 1)$. By Sudakov's minoration Fact H.9, we have

$$\sqrt{\log\left| \mathcal{N}_{\varepsilon, \|\cdot\|_{\mathrm{F}}}(\mathcal{L}_s) \right|} \leqslant \frac{2}{\varepsilon} \, \mathbb{E} \sup_{L \in \mathcal{L}_s} \langle W, L \rangle \qquad\qquad \text{(Fact H.9)}$$

$$\leqslant \frac{2}{\varepsilon} \sup_{L \in \mathcal{L}_s} \mathbb{E}\|W\| \cdot \|L\|_{\mathrm{nuc}} \qquad\qquad \text{(Hölder's inequality)}$$

$$\leqslant \frac{2s}{\varepsilon} \, \mathbb{E}\|W\| \qquad\qquad \text{(Definition of } \mathcal{L}_s)$$

$$\leqslant \frac{4s\sqrt{n}}{\varepsilon} \qquad\qquad \text{(Fact H.6)},$$

where in the last inequality we use a bound on the expected spectral norm of a Gaussian matrix Fact H.6. $\qquad\square$

Hence the intersection of the set $\mathcal{S}_4(\overline{\Omega}) = \left\{ L \in \mathbb{R}^{n \times n} \;\middle|\; \|L\|_{\mathrm{nuc}} \leqslant 4\|L_{\overline{\Omega}}\|_{\mathrm{nuc}} \right\}$ with the ball $\{L \in \mathbb{R}^{n \times n} \mid \|L\|_{\mathrm{F}} \leqslant 1\}$ has $\varepsilon$-net of size $\exp\left[ \frac{16 \cdot 32 \cdot n}{\varepsilon^2} \right] \leqslant \exp\left[ \frac{600 \cdot n}{\varepsilon^2} \right]$.

Now we can prove the restricted local strong convexity:

**Lemma C.3** (Restricted local strong convexity of Huber-loss). *Consider the settings of Theorem 2.1. Let* $0 < \delta < 1, R > 0$ *and* $h \geqslant \rho/n + \zeta$.

*Define*

$$\mathcal{B}_R := \left\{ \Delta \in \mathbb{R}^{n \times n} \;\middle|\; \|\Delta\|_{\mathrm{F}} = R, \|L^* + \Delta\|_{\max} \leqslant \rho/n \right\}.$$

*Suppose that*

$$R \geqslant 2000 \cdot \frac{\rho/n}{\alpha} \cdot \sqrt{rn + \log(2/\delta)}.$$

*Then with probability at least* $1 - \delta/2$, *for all* $\Delta \in \mathcal{B}_R \cap \mathcal{S}_4(\overline{\Omega})$,

$$F_h(L^* + \Delta) \geqslant F_h(L^*) + \langle \nabla F_h(L^*), \Delta \rangle + 0.01 \cdot \alpha \cdot \|\Delta\|_{\mathrm{F}}^2.$$

*Proof.* Denote $M = \rho/n$. Consider $L$ such that $\|L\|_{\max} \leqslant M$. Since $h \geqslant \zeta + M$, by Lemma G.2,

$$F_h(L) - F_h(L^*) - \langle \nabla F_h(L^*), L - L^* \rangle$$

$$\geqslant \frac{1}{2} \sum_{i,j \in [n]} (L_{ij} - L_{ij}^*)^2 \, \mathbf{1}_{\left[ \left| L_{ij}^* \right| \leqslant h - \zeta \right]} \cdot \mathbf{1}_{\left[ \left| L_{ij} - L_{ij}^* \right| \leqslant \zeta \right]} \qquad\qquad \text{(Lemma G.2)}$$

$$= \frac{1}{2} \sum_{i,j \in [n]} (L_{ij} - L_{ij}^*)^2 \, \mathbf{1}_{\left[ |N_{ij}| \leqslant \zeta \right]}. \qquad\qquad (\|L^*\|_{\max} \leqslant M \leqslant h - \zeta \;\&\; L_{ij} - L_{ij}^* = N_{ij})$$

We will lower bound this quantity for every $L$ such that $L - L^* \in \mathcal{B}_R \cap \mathcal{S}_4(\overline{\Omega})$. Denote $\mathcal{C}_R := \mathcal{B}_R \cap \mathcal{S}_4(\overline{\Omega})$ and let $\Delta := L - L^* \in \mathcal{C}_R$. By Lemma C.2, there exists $(\varepsilon \cdot R)$-net $\mathcal{N}_{\varepsilon R, \|\cdot\|_{\mathrm{F}}}(\mathcal{C}_R)$ of size at most $\exp\left[ \frac{16 \cdot 32 \cdot n}{\varepsilon^2} \right] \leqslant \exp\left[ \frac{600 \cdot n}{\varepsilon^2} \right]$. (recall that $s^2 = 32r$). Thus, we can write $\Delta \in \mathcal{C}_R$ as a sum $A + B \in \mathbb{R}^{n \times n}$ where $A \in \mathcal{N}_{\varepsilon R, \|\cdot\|_{\mathrm{F}}}(\mathcal{C}_R)$ and $\|B\|_{\mathrm{F}} \leqslant \varepsilon R$. It follows that

$$\sum_{i,j \in [n]} \Delta_{ij}^2 \cdot \mathbf{1}_{\left[ |N_{ij}| \leqslant \zeta \right]} = \sum_{i,j \in [n]} (A_{ij} + B_{ij})^2 \cdot \mathbf{1}_{\left[ |N_{ij}| \leqslant \zeta \right]}$$

$$\geqslant \frac{1}{2} \sum_{i,j\in[n]} A_{ij}^2 \cdot \mathbf{1}_{[|N_{ij}|\leqslant\zeta]} - \sum_{i,j\in[n]} B_{ij}^2 \cdot \mathbf{1}_{[|N_{ij}|\leqslant\zeta]} . \qquad \text{(C.4)}$$

Let $\varepsilon = \sqrt{\alpha}/4$. Then

$$\sum_{i,j\in[n]} B_{ij}^2 \cdot \mathbf{1}_{[|N_{ij}|\leqslant\zeta]} \leqslant \|B\|_F^2 \leqslant \varepsilon^2 R^2 \leqslant \frac{\alpha \cdot R^2}{16} . \qquad \text{(C.5)}$$

Denote $E := \mathbb{E} \sum_{i,j\in[n]} A_{ij}^2 \cdot \mathbf{1}_{[|N_{ij}|\leqslant\zeta]}$. Since $A \in \mathcal{N}_{\varepsilon R, \|\cdot\|_F}(C_R) \subset C_R$, we have $\|A\|_F = R$, hence

$$E \geqslant \alpha\|A\|_F^2 \geqslant \frac{\alpha \cdot R^2}{2} . \qquad \text{(C.6)}$$

Moreover, since $\|A\|_{\max} \leqslant \|-L^*\|_{\max} + \|A + L^*\|_{\max} \leqslant 2M$ (recall $A \in \mathcal{B}_R$) and $\alpha_{ij} = \mathbb{P}\big(|N_{ij}| \leqslant \zeta\big)$, we have $\big|A_{ij}^2\big(\mathbf{1}_{[|N_{ij}|\leqslant\zeta]} - \alpha_{ij}\big)\big| \leqslant 4M^2$, implying that

$$\sum_{i,j\in[n]} \mathbb{E} A_{ij}^4 \big(\mathbf{1}_{[|N_{ij}|\leqslant\zeta]} - \alpha_{ij}\big)^2 \leqslant 4M^2 \sum_{i,j\in[n]} \mathbb{E} A_{ij}^2 \big|\mathbf{1}_{[|N_{ij}|\leqslant\zeta]} - \alpha_{ij}\big|$$

$$= 4M^2 \sum_{i,j\in[n]} \Big(\alpha_{ij} \cdot A_{ij}^2 \cdot 0 + (1 - \alpha_{ij}) \cdot A_{ij}^2 \cdot \alpha_{ij}\Big)$$

$$= 4M^2 \sum_{i,j\in[n]} A_{ij}^2 \cdot (\alpha_{ij} - \alpha_{ij}^2)$$

$$\leqslant 4M^2 E .$$

Applying Bernstein's inequality ([Fact H.3](#)) with $t \geqslant 1$ we get

$$\mathbb{P}\left(\left|\sum_{i,j\in[n]} A_{ij}^2 \big(\mathbf{1}_{[|N_{ij}|\leqslant\zeta]} - \alpha_{ij}\big)\right| \geqslant t \cdot 2M \cdot \sqrt{E} + t^2 \cdot 4M^2\right) \leqslant 2\exp(-t^2/4) .$$

Note that $\big|\mathcal{N}_{\varepsilon R, \|\cdot\|_F}(C_R)\big| \leqslant \exp\big[\frac{600rn}{\varepsilon^2}\big] \leqslant \exp\big[\frac{10000rn}{\alpha}\big]$. Therefore, if we set

$$t = \sqrt{\frac{40000rn}{\alpha} + 8\log(2/\delta)}$$

and take a union bound over $\mathcal{N}_{\varepsilon R, \|\cdot\|_F}(C_R)$, we obtain that with probability at least $1 - \delta/2$, we have

$$\left|\sum_{i,j\in[n]} A_{ij}^2 \cdot \big(\mathbf{1}_{[|N_{ij}|\leqslant\zeta]} - \alpha_{ij}\big)\right| \leqslant 400M \cdot \sqrt{E} \cdot \sqrt{\frac{rn}{\alpha} + \log(2/\delta)} + (400M)^2\Big(\frac{rn}{\alpha} + \log(2/\delta)\Big) .$$

for all $A \in \mathcal{N}_{\varepsilon R, \|\cdot\|_F}(C_R)$. Now since $E \geqslant \frac{\alpha R^2}{2}$ and $R \geqslant \frac{1000M}{\sqrt{\alpha}}\sqrt{\frac{rn}{\alpha} + \log(2/\delta)}$, we have

$$\sqrt{E} \geqslant \frac{\sqrt{\alpha} \cdot R}{\sqrt{2}} \geqslant 1400M\sqrt{\frac{rn}{\alpha} + \log(2/\delta)},$$

hence, with probability at least $1 - \delta/2$,

$$\left|\sum_{i,j\in[n]} A_{ij}^2 \cdot \big(\mathbf{1}_{[|N_{ij}|\leqslant\zeta]} - \alpha_{ij}\big)\right| \leqslant \frac{2}{7} \cdot E + \Big(\frac{2}{7}\Big)^2 \cdot E \leqslant 4E .$$

By combining this with [Eq. (C.4)](#), [Eq. (C.5)](#) and [Eq. (C.6)](#), we obtain that with probability at least $1 - \delta$, we have

$$\sum_{i,j\in[n]} \Delta_{ij}^2 \cdot \mathbf{1}_{[|N_{ij}|\leqslant 1]} \geqslant \frac{1}{2}(E - 0.4E) - \frac{\alpha \cdot R^2}{16} \geqslant 0.15\alpha \cdot R^2 - 0.0625\alpha \cdot R^2 \geqslant 0.08\alpha R^2$$

concluding the proof. $\qquad\qquad \square$

**Putting everything together.** We can now combine the above results with [Theorem B.1](#) to prove [Theorem 2.1](#).

***Proof of [Theorem 2.1](#).*** By [Lemma C.1](#) and [Lemma C.3](#), we can apply [Theorem B.1](#) with $\gamma = 100\left(\zeta + \frac{\rho}{n}\right)\sqrt{n + \log(2/\delta)}$, $\kappa = 0.01\alpha$, and $s = 4\sqrt{2r}$.

It follows that for

$$R \gtrsim (\zeta + \rho/n)\sqrt{\frac{r(n + \log(2/\delta))}{\alpha^2}}$$

the estimator $\hat{L}$ defined in [Eq. (C.1)](#) satisfies $\left\|\hat{L} - L^*\right\|_{\mathrm{F}} < R$ with probability at least $1 - \delta$. With $\delta = 2^{-n}$ we get the desired bound. $\qquad\square$

## D  Sparse linear regression with oblivious outliers ([Theorem 2.2](#))

We prove [Theorem 2.2](#), which will be restated below. Before the restatement, for easier reference, we list the three assumptions in [Section 2.2](#) for the design matrix $X \in \mathbb{R}^{n \times n}$:

1. For every column $X^i$ of $X$, $\left\|X^i\right\| \leqslant \sqrt{\nu n}$.

2. *Restricted eigenvalue property (RE-property):* For every vector $u \in \mathbb{R}^d$ such that[12] $\left\|u_{\mathrm{supp}(\beta^*)}\right\|_1 \geqslant 0.1 \cdot \|u\|_1$, $\frac{1}{n}\|Xu\|^2 \geqslant \lambda \cdot \|u\|^2$ for some parameter $\lambda > 0$.

3. *Well-spreadness property:* For some $m \in [n]$ and for every vector $u \in \mathbb{R}^d$ such that $\left\|u_{\mathrm{supp}(\beta^*)}\right\|_1 \geqslant 0.1 \cdot \|u\|_1$ and for every subset $S \subseteq [n]$ with $|S| \geqslant n - m$, it holds that $\|(Xu)_S\| \geqslant \frac{1}{2}\|Xu\|$.

Recall that $F_2(\beta) = \sum_{i=1}^n f_2\left(y_i - \langle X_i, \beta \rangle\right)$, where

$$f_2(t) := \begin{cases} \frac{1}{2}t^2 & \text{for } |t| \leqslant 2\,, \\ 2|t| - 2 & \text{otherwise.} \end{cases}$$

**Theorem D.1** (Restatement of [Theorem 2.2](#)). *Let $\beta^* \in \mathbb{R}^d$ be an unknown $k$-sparse vector and let $X \in \mathbb{R}^{n \times d}$ be a deterministic matrix such that for each column $X^i$ of $X$, $\|X^i\| \leqslant \sqrt{\nu n}$, satisfying the RE-property with $\lambda > 0$ and well-spreadness property with $m \gtrsim \frac{k \log d}{\lambda \cdot \alpha^2}$ (recall that $n \geqslant m$). Further, let $\eta$ be an $n$-dimensional random vector with independent, symmetrically distributed (about zero) entries and $\alpha = \min_{i \in [n]} \mathbb{P}\left\{\left|\eta_i\right| \leqslant 1\right\}$. Consider the following estimator:*

$$\hat{\beta} := \arg\min_{\beta \in \mathbb{R}^d}\left(F_2(\beta) + 100\sqrt{\nu n \log d} \cdot \left\|\beta\right\|_1\right). \tag{D.1}$$

*Then, with probability at least $1 - d^{-10}$ over $\eta$, given $X$ and $y = X\beta^* + \eta$, the estimator $\hat{\beta}$ satisfies*

$$\frac{1}{n}\left\|X\left(\hat{\beta} - \beta^*\right)\right\|^2 \leqslant O\left(\frac{\nu}{\lambda} \cdot \frac{k \log d}{\alpha^2 \cdot n}\right) \qquad \text{and} \qquad \left\|\hat{\beta} - \beta^*\right\|^2 \leqslant O\left(\frac{\nu}{\lambda^2} \cdot \frac{k \log d}{\alpha^2 \cdot n}\right).$$

We assume $d \geqslant 2$ since for $d = 1$ [Theorem D.1](#) is trivially true (since the probability $1 - d^{-10} = 0$ in this case).

As for principal component analysis ([Appendix C](#)), we will prove [Theorem D.1](#) by showing that the estimator [Eq. (D.1)](#) fulfills the conditions of [Theorem B.1](#) with $F(\beta) = F_2(\beta)$, $\|u\|_{\mathrm{reg}} = \|u\|_1$, $\gamma = 100\sqrt{n \log d}$ and $\mathcal{E}(u) = \frac{1}{\sqrt{n}}\|Xu\|$.

Let $\Omega := \{\beta \in \mathbb{R}^d \mid \mathrm{supp}\,\beta \subseteq \mathrm{supp}(\beta^*)\}$ and $\overline{\Omega} := \Omega$. Clearly, for any $v \in \Omega$ and any $v' \in \overline{\Omega}^\perp$,

$$\|v + v'\|_1 = \|v\|_1 + \|v'\|_1\,.$$

---

[12]For a vector $v \in \mathbb{R}^d$ and a set $S \subseteq [d]$, we denote by $v_S$ the restriction of $v$ to the coordinates in $S$.

That is, $\|\cdot\|_1$ is decomposable, satisfying condition Eq. (B.1).

The contraction condition Eq. (B.2) holds for $s = 4\sqrt{k/\lambda}$ since for all $v \in \mathcal{S}_4(\overline{\Omega}) = \left\{ v \mid \|v\|_1 \leqslant 4\|v_{\overline{\Omega}}\|_1 \right\}$, $\|v\|_1 \leqslant 4\sqrt{k}\|v\| \leqslant 4\sqrt{k/\lambda} \cdot \frac{1}{\sqrt{n}}\|Xv\|$, where the last inequality comes from the RE-property.

It remains to provide a gradient bound of the form in Eq. (B.3) and local strong convexity in Eq. (B.4).

**Lemma D.2** (Gradient bound). *Consider the settings of Theorem D.1. Then, with probability at least* $1 - \delta/2$,

$$\left\|\nabla F_2(\beta^*)\right\|_{\max} \leqslant 20\sqrt{v \cdot n \cdot (\log d + \log(2/\delta))}.$$

*Proof.* By definition of $f_2$,

$$\nabla\left(\sum_{i=1}^{n} f_2(y_i - \langle X_i, \beta^*\rangle)\right) = z^\top X$$

where $z$ is a $n$-dimensional random vector with independent, symmetric entries $f_2'(\eta_i)$ bounded by 2 in absolute value. By Hoeffding's inequality (Fact H.2), for $t \geqslant 0$,

$$\mathbb{P}(|\langle z, X_i\rangle| \geqslant 10t \cdot 2 \cdot \|X_i\|) \leqslant \exp(-t^2).$$

Since $\|X_i\| \leqslant \sqrt{vn}$, taking a union bound over all $j \in [d]$ yields the statement. $\qquad\square$

**Proof of local strong convexity.** We first bound the size of an $\varepsilon$-net for the set of approximately sparse vectors (Lemma D.3) and then prove the required local strong convexity bound (Lemma D.4).

**Lemma D.3** ($\varepsilon$-Net for approximately sparse vectors). *Let* $0 < \varepsilon < 1$ *and*

$$\mathcal{U}_s := \left\{ \beta \in \mathbb{R}^d \;\middle|\; \|\beta\|_1 \leqslant s \cdot \frac{1}{\sqrt{n}}\|X\beta\|, \frac{1}{\sqrt{n}}\|X\beta\| \leqslant 1 \right\}.$$

*Then* $\mathcal{U}_s$ *has an* $\varepsilon$*-net of size* $\exp\left[\frac{16s^2 v \log d}{\varepsilon^2}\right]$ *in terms of distance* $\rho(\beta, \beta') := \frac{1}{\sqrt{n}}\|X(\beta - \beta')\|$.

*Proof.* Let $w$ be an $n$-dimensional random Gaussian vector $w \sim N(0, \mathrm{Id}_n)$. By Sudakov's minoration (Fact H.9), for

$$\sqrt{\log|\mathcal{N}_{\varepsilon, \rho(\cdot, \cdot)}(\mathcal{U}_s)|} \leqslant \frac{2}{\varepsilon}\, \mathbb{E}\, \frac{1}{\sqrt{n}}\, \sup_{\beta \in \mathcal{U}_s} \langle w, X\beta\rangle \qquad\qquad \text{(Fact H.9)}$$

$$= \frac{2}{\varepsilon}\, \mathbb{E}\, \frac{1}{\sqrt{n}}\, \sup_{\beta \in \mathcal{U}_s} \langle X^\top w, \beta\rangle$$

$$\leqslant \frac{2}{\varepsilon}\, \mathbb{E}\, \frac{1}{\sqrt{n}}\, \sup_{\beta \in \mathcal{U}_s} \left\|X^\top w\right\|_{\max} \|\beta\|_1 \qquad \text{(Hölder's inequality)}$$

$$\leqslant \frac{2s}{\varepsilon}\, \mathbb{E}\, \frac{1}{\sqrt{n}} \left\|X^\top w\right\|_{\max} \qquad\qquad \text{(Definition of } \mathcal{U}_s\text{)},$$

$$\leqslant \frac{4s\sqrt{v \log d}}{\varepsilon},$$

where in the last inequality we use the bound on the expected maximal entry of a vector with $v$-subgaussian entries Fact H.4. $\qquad\square$

**Lemma D.4** (Restricted local strong convexity of Huber-loss). *Consider the settings of Theorem D.1. Let* $0 < \delta < 1, R > 0$. *Define*

$$\mathcal{B}_R := \left\{ u \in \mathbb{R}^d \;\middle|\; \frac{1}{\sqrt{n}}\|Xu\| = R \right\}.$$

*Suppose that the set size $m$ from the well-spread property satisfies $m \geqslant 4R^2 n$ and*

$$R \geqslant 100 \cdot \sqrt{\frac{\nu k \log d + \log(2/\delta)}{\lambda \cdot \alpha^2 \cdot n}} \, .$$

*Then with probability at least $1 - \delta/2$, for all $u \in \mathcal{B}_R \cap \mathcal{S}_4(\overline{\Omega})$,*

$$F_2(\beta) \geqslant F_2(y - X\beta^*) + \langle \nabla F_2(\beta^*), u \rangle + 0.01 \cdot \alpha n \cdot \frac{1}{n} \|Xu\|^2 \, .$$

*Proof.* Denote $C_R = \mathcal{B}_R \cap \mathcal{S}_4(\overline{\Omega})$ and let $u \in C_R$. By [Lemma G.2](#),

$$F_2(\beta^* + u) - F_2(\beta^*) - \langle \nabla F_2(\beta^*), u \rangle \geqslant \frac{1}{2} \sum_{i \in [n]} \langle X_i, u \rangle^2 \mathbf{1}_{[|\eta_i| \leqslant 1]} \cdot \mathbf{1}_{[|\langle X_i, u \rangle| \leqslant 1]} \, .$$

Note that for any $u \in \mathcal{B}_R$ there are at most $4R^2 n$ coordinates of $Xu$ larger than $1/4$ in absolute value, and since $X$ is well-spread for sets of size $m = 4R^2 n$,

$$\sum_{i \in [n]} \langle X_i, u \rangle^2 \mathbf{1}_{[\langle X_i, u \rangle^2 \leqslant 1/4]} \geqslant \frac{1}{4} \|Xu\|^2 \, .$$

Thus

$$E := \mathbb{E} \sum_{i \in [n]} \langle X_i, u \rangle^2 \mathbf{1}_{[|\eta_i| \leqslant 1]} \cdot \mathbf{1}_{[\langle X_i, u \rangle^2 \leqslant 1/4]} \geqslant \frac{1}{4} \cdot \alpha \cdot \|Xu\|^2 = \frac{\alpha R^2 n}{4}$$

We now bound the deviation. We have

$$\text{for all } i \in [n], \quad \langle X_i, u \rangle^2 \mathbf{1}_{[|\eta_i| \leqslant 1]} \cdot \mathbf{1}_{[\langle X_i, u \rangle^2 \leqslant 1/4]} \leqslant 1$$

$$\text{and} \quad \mathbb{E} \sum_{i \in [n]} \left[ \langle X_i, u \rangle^2 \cdot \mathbf{1}_{[\langle X_i, u \rangle^2 \leqslant 1/4]} \cdot \left( \mathbf{1}_{[|\eta_i| \leqslant 1]} - \alpha_i \right) \right]^2$$

$$\leqslant \mathbb{E} \sum_{i \in [n]} \langle X_i, u \rangle^2 \cdot \mathbf{1}_{[\langle X_i, u \rangle^2 \leqslant 1/4]} \cdot \mathbf{1}_{[|\eta_i| \leqslant 1]}$$

$$\leqslant E \, .$$

Applying Bernstein's inequality [Fact H.3](#)

$$\mathbb{P}\left( \sum_{i \in [n]} \langle X_i, u \rangle^2 \cdot \mathbf{1}_{[\langle X_i, u \rangle^2 \leqslant 1/4]} \cdot \left( \mathbf{1}_{[|\eta_i| \leqslant 1]} - \alpha_i \right) \geqslant t \cdot \sqrt{E} + t^2 \right) \leqslant \exp\{-t^2/4\} \, .$$

It remains to extend uniformly this bound over all $u \in C_R$. By [Lemma D.3](#) there exists an $(\varepsilon \cdot R)$-net $\mathcal{N}_{\varepsilon R}(C_R)$ of size $\exp\left[ \frac{256 \nu k \log d}{\lambda \varepsilon^2} \right]$ (recall that $s = 4\sqrt{k/\lambda}$). Thus for any $u \in C_R$ there exists $u' \in \mathcal{N}_{\varepsilon R}(C_R)$ such that $\frac{1}{\sqrt{n}} \|X(u - u')\| \leqslant \varepsilon R$ and consequently

$$\sum_{i \in [n]} \langle X_i, u' \rangle^2 \mathbf{1}_{[|\eta_i| \leqslant 1]} \cdot \mathbf{1}_{[\langle X_i, u' \rangle^2 \leqslant 1/4]} \leqslant \sum_{i \in [n]} \langle X_i, u' \rangle^2 \mathbf{1}_{[|\eta_i| \leqslant 1]} \cdot \mathbf{1}_{[\langle X_i, u \rangle^2 \leqslant 1]} \mathbf{1}_{[\langle X_i, u' \rangle^2 \leqslant 1/4]} + \varepsilon^2 R^2 n$$

$$\leqslant 2 \sum_{i \in [n]} \langle X_i, u \rangle^2 \mathbf{1}_{[|\eta_i| \leqslant 1]} \cdot \mathbf{1}_{[\langle X_i, u \rangle^2 \leqslant 1]} \cdot \mathbf{1}_{[\langle X_i, u' \rangle^2 \leqslant 1/4]} + \varepsilon^2 R^2 n$$

$$+ 2 \sum_{i \in [n]} \langle X_i, u' - u \rangle^2 \mathbf{1}_{[|\eta_i| \leqslant 1]} \cdot \mathbf{1}_{[\langle X_i, u \rangle^2 \leqslant 1]} \cdot \mathbf{1}_{[\langle X_i, u' \rangle^2 \leqslant 1/4]}$$

$$\leqslant 2 \sum_{i \in [n]} \langle X_i, u \rangle^2 \mathbf{1}_{[|\eta_i| \leqslant 1]} \cdot \mathbf{1}_{[\langle X_i, u \rangle^2 \leqslant 1]} + 3\varepsilon^2 R^2 n \, .$$

The first inequality holds since each term at the first sum that doesn't appear in the second sum corresponds to the index $i \in [n]$ such that $\langle X_i, u - u' \rangle^2 \geqslant 1/4$, and since each term is bounded by $1/4$, their sum is bounded by $\sum_{i \in [n]} \langle X_i, u - u' \rangle^2 \leqslant \varepsilon^2 R^2 n$.

Setting $\varepsilon = \sqrt{\alpha}/4$ and taking a union bound, with probability at least $1 - \delta/2$ for all unit vectors $u \in \mathcal{L}_{k,R}$ we get

$$\sum_{i \in [n]} \langle X_i, u \rangle^2 \mathbf{1}_{[|\eta_i| \leqslant 1]} \mathbf{1}_{[\langle X_i, u \rangle^2 \leqslant 4]} \geqslant \frac{E}{2} - \frac{3\varepsilon^2 R^2 n}{2} - \sqrt{E} \frac{\sqrt{64\nu k \log d + 4\log(\frac{2}{\delta})}}{\sqrt{\lambda}\varepsilon} - \frac{128\nu k \log d + 4\log(\frac{2}{\delta})}{\lambda \varepsilon^2}$$

$$\geqslant 0.01 \cdot \alpha \cdot R^2 n \,.$$

$\square$

**Putting things together.** We combine the above results with Theorem B.1.

***Proof of Theorem 2.2.*** By Lemma D.2 and Lemma D.4, we can apply Theorem B.1 with $\gamma = 100\sqrt{\nu \cdot n(\log d + \log(2/\delta))}$, $\kappa = 0.01 \cdot \alpha \cdot n$ and $s = 4\sqrt{k/\lambda}$. It follows that for

$$R \gtrsim \sqrt{\frac{\nu \cdot k \cdot (\log d + \log(2/\delta))}{\lambda \cdot \alpha^2 \cdot n}} \,,$$

the estimator $\hat{\beta}$ defined in Eq. (D.1) with probability $1 - \delta$ satisfies $\frac{1}{n}\left\| X\left(\hat{\beta} - \beta^*\right) \right\| \leqslant R$. Taking $\delta = d^{-10}$, we get the desired bound. Since $\hat{\beta} - \beta \in \mathcal{S}_4(\Omega)$, we also get the desired parameter error $\left\| \hat{\beta} - \beta \right\| \leqslant R/\sqrt{\lambda}$. $\square$

# E  Sparse linear regression with Gaussian design (Theorem 2.3)

In this section we will prove Theorem 2.3. As before, we will use Theorem B.1. Recall that in this setting, our model looks as follows:

$$y = X\beta^* + \eta$$

where $X \in \mathbb{R}^{n \times d}$ is a *random* matrix whose rows $X_1, \ldots, X_n$ are i.i.d. $N(0, \Sigma)$ and $\eta \in \mathbb{R}^n$ is a *deterministic* vector such that $\alpha n$ coordinates have absolute value bounded by 1. We restate Theorem 2.3 here for completeness:

**Theorem E.1** (Restatement of Theorem 2.3)**.** *Let $\beta^* \in \mathbb{R}^d$ be an unknown $k$-sparse vector and let $X$ be a $n$-by-$d$ random matrix with i.i.d. rows $X_1, \ldots X_n \sim N(0, \Sigma)$ for a positive definite matrix $\Sigma$. Further, let $\eta \in \mathbb{R}^n$ be a deterministic vector with $\alpha \cdot n$ coordinates bounded by 1 in absolute value. Suppose that $n \gtrsim \frac{\nu(\Sigma) \cdot k \log d}{\sigma_{\min}(\Sigma) \cdot \alpha^2}$, where $\nu(\Sigma)$ is the maximum diagonal entry of $\Sigma$ and $\sigma_{\min}(\Sigma)$ is its smallest eigenvalue. Then, with probability at least $1 - d^{-10}$ over $X$, given $X$ and $y = X\beta^* + \eta$, the estimator Eq. (2.3) satisfies*

$$\frac{1}{n}\left\| X\left(\hat{\beta} - \beta^*\right) \right\|^2 \leqslant O\left(\frac{\nu(\Sigma) \cdot k \log d}{\sigma_{\min}(\Sigma) \cdot \alpha^2 \cdot n}\right) \qquad and \qquad \left\| \hat{\beta} - \beta^* \right\|^2 \leqslant O\left(\frac{\nu(\Sigma) \cdot k \log d}{\sigma_{\min}^2(\Sigma) \cdot \alpha^2 \cdot n}\right).$$

As in the previous section, we assume $d \geqslant 2$ since for $d = 1$ Theorem 2.3 is true (since the probability $1 - d^{-10} = 0$ in this case).

First, we bound the gradient of Huber loss. Then, to prove restricted local strong convexity of Huber loss, we show that the values of the empirical covariance (as a quadratic form) on approximately $k$-sparse vectors are well-concentrated near the values of the actual covariance. The proof first appeared in (RWY10), but they only stated the result in terms of a lower bound on the values of empirical covariance and did not discuss an upper bound, though the proof of the upper bound is very similar. Then we use this concentration to prove well-spreadness and restricted local strong convexity.

Recall that $F_2(\beta) = \sum_{i=1}^n f_2\left(y_i - \langle X_i, \beta \rangle\right)$, where

$$f_2(t) := \begin{cases} \frac{1}{2}t^2 & \text{for } |t| \leqslant 2 \,, \\ 2|t| - 2 & \text{otherwise.} \end{cases}$$

**Gradient bound for Gaussian design.**

**Lemma E.2.** *Consider the settings of Theorem 2.3. Then with probability at least $1 - \delta/2$*

$$\left\|\nabla F_2(\beta^*)\right\|_{\max} \leqslant 4\sqrt{\nu(\Sigma) \cdot n \cdot (\log d + \log(2/\delta))}\,.$$

*Proof.* By definition of the Huber loss and choice of the Huber penalty

$$\nabla F_2(\beta^*) = \nabla\left(\sum_{i=1}^{n} f_2(y_i - \langle X_i, \beta^*\rangle)\right) = z^{\top}X$$

where $z$ is an $n$-dimensional vector whose entries $f_2'(\eta_i)$ are bounded by 2 in absolute value. Since $\frac{1}{\|z\|}\Sigma^{-1/2}X^{\top}z = g \sim N(0,1)^n$,

$$\left\|z^{\top}X\right\| = \|z\| \cdot \left\|\Sigma^{1/2}g\right\| \leqslant 2\sqrt{n} \cdot \sqrt{\nu(\Sigma) \cdot (2\log d + 4\log(2/\delta))}\,,$$

where we used the union bound over all $j \in [d]$ and the standard tail bounds for Gaussian variables $\left(\Sigma^{1/2}g\right)_j$ whose variance is $\Sigma_{jj}$. $\qquad\square$

**Concentration of empirical covariance on approximately $k$-sparse vectors.** To prove well-spreadness and restricted local strong convexity in case of Gaussian design $X$, we will need the fact that for all approximately $k$-sparse vectors $u$, $\frac{1}{n}\|Xu\|^2 \approx \left\|\Sigma^{1/2}u\right\|^2$ as long as $n \gtrsim \frac{\nu(\Sigma)k\log d}{\sigma_{\min}(\Sigma)}$. Formally, we will use the following theorem:

**Theorem E.3.** *Let $X$ be a $n$-by-$d$ random matrix with i.i.d. rows $X_1, \ldots X_n \sim N(0, \Sigma)$, where $\Sigma$ is a positive definite matrix. Suppose that for some $K \geqslant 1$, $n \geqslant 1000 \cdot \frac{\nu(\Sigma)}{\sigma_{\min}(\Sigma)} \cdot K\log d$. Then with probability at least $1 - \exp(-n/100)$, for all $u \in \mathbb{R}^d$ such that $\|u\|_1 \leqslant \sqrt{K}\|u\|$,*

$$\frac{1}{2}\left\|\Sigma^{1/2}u\right\| \leqslant \frac{1}{\sqrt{n}}\|Xu\| \leqslant 2\left\|\Sigma^{1/2}u\right\|\,. \tag{E.1}$$

The first inequality of Theorem E.3 was shown in (RWY10) (see also (Wai19), section 7.3.3), and the second inequality can be proved in a very similar way. For completeness, we provide a proof of second inequality.

*Proof of the second inequality of Theorem E.3.* Since the inequality is scale invariant, it is enough to show it for $u \in \mathbb{R}^d$ such that $\left\|\Sigma^{1/2}u\right\| = 1$. For $s > 0$ denote

$$\mathcal{U}_s := \left\{u \in \mathbb{R}^d \mid \left\|\Sigma^{1/2}u\right\| = 1, \|u\|_1 \leqslant s\right\} \qquad \text{and} \qquad \mathcal{M}_s(X) := \sup_{u \in \mathcal{U}_s} \frac{1}{\sqrt{n}}\|Xu\|\,.$$

First, we bound the expectation of $\mathcal{M}_s(X)$:

**Lemma E.4.**

$$\mathbb{E}\,\mathcal{M}_s(X) \leqslant 1 + 2s\sqrt{\frac{\nu(\Sigma)\log d}{n}}\,.$$

*Proof.* Consider Gaussian process $W_{u,v} = v^{\top}Xu$ for $(u,v) \in \mathcal{U}_s \times S^{n-1}$, where $S^{n-1}$ is a unit sphere in $\mathbb{R}^n$. Denote $\mathcal{P} = \mathcal{U}_s \times S^{n-1}$. Our goal is to bound $\frac{1}{\sqrt{n}}\mathbb{E}\sup_{(u,v)\in\mathcal{P}} W_{u,v}$.

Denote $G = X\Sigma^{-1/2}$. For all $(u,v), (\tilde{u}, \tilde{v}) \in \mathcal{P}$,

$$\mathbb{E}(W_{u,v} - W_{\tilde{u},\tilde{v}})^2 = \mathbb{E}\langle X^{\top}, uv^{\top} - \tilde{u}\tilde{v}^{\top}\rangle^2 = \mathbb{E}\langle G^{\top}, \Sigma^{1/2}uv^{\top} - \Sigma^{1/2}\tilde{u}\tilde{v}^{\top}\rangle^2 = \left\|\Sigma^{1/2}uv^{\top} - \Sigma^{1/2}\tilde{u}\tilde{v}^{\top}\right\|_{\mathrm{F}}^2\,.$$

Now consider another Gaussian process $Z_{u,v} = g^{\top}\Sigma^{1/2}u + h^{\top}v$, where $g \sim N(0, \mathrm{Id}_d)$ and $h \sim N(0, \mathrm{Id}_n)$. For all $(v,u), (\tilde{v}, \tilde{u}) \in \mathcal{P}$,

$$\mathbb{E}(Z_{u,v} - Z_{\tilde{u},\tilde{v}})^2 = \mathbb{E}\langle g, \Sigma^{1/2}(u - \tilde{u})\rangle^2 + \mathbb{E}\langle h, v - \tilde{v}\rangle^2 = \left\|\Sigma^{1/2}u - \Sigma^{1/2}\tilde{u}\right\|^2 + \|v - \tilde{v}\|^2\,.$$

Note that for every quadruple of unit vectors $x, \tilde{x} \in \mathbb{R}^d$, $y, \tilde{y} \in \mathbb{R}^n$,

$$
\begin{aligned}
\|xy^\mathsf{T} - \tilde{x}\tilde{y}^\mathsf{T}\|_\mathrm{F}^2 &= \|(x - \tilde{x})y^\mathsf{T} + \tilde{x}(y^\mathsf{T} - \tilde{y}^\mathsf{T})\|_\mathrm{F}^2 \\
&= \|y\|^2 \|x - \tilde{x}\|^2 + \|\tilde{x}\|^2 \|y - \tilde{y}\|^2 + 2\operatorname{Tr} y(x - \tilde{x})^\mathsf{T} \tilde{x}(y^\mathsf{T} - \tilde{y}^\mathsf{T}) \\
&= \|x - \tilde{x}\|^2 + \|y - \tilde{y}\|^2 + 2\big(\langle x, \tilde{x}\rangle - \|\tilde{x}\|^2\big) \cdot \big(\|y\|^2 - \langle y, \tilde{y}\rangle\big) \\
&\leqslant \|x - \tilde{x}\|^2 + \|y - \tilde{y}\|^2 \,.
\end{aligned}
$$

Hence for all $(u, v), (\tilde{u}, \tilde{v}) \in \mathcal{P}$, $\mathbb{E}(W_{u,v} - W_{\tilde{u},\tilde{v}})^2 \leqslant \mathbb{E}(Z_{u,v} - Z_{\tilde{u},\tilde{v}})^2$, and by Sudakov–Fernique theorem Fact H.8,

$$
\mathbb{E} \sup_{(u,v)\in\mathcal{P}} W_{u,v} \leqslant \mathbb{E} \sup_{(u,v)\in\mathcal{P}} Z_{u,v} \,.
$$

Therefore, it is enough to bound $\mathbb{E} \sup_{u\in\mathcal{U}_s} g^\mathsf{T}\Sigma^{1/2}u + \mathbb{E} \sup_{\|v\|=1} h^\mathsf{T}v$. The second term is just an expectation of $\chi$ distributed variable, and can be bounded using Jensen's inequality:

$$
\mathbb{E} \sup_{\|v\|=1} h^\mathsf{T}v = \mathbb{E}\|h\| \leqslant \sqrt{\mathbb{E}\|h\|^2} \leqslant \sqrt{n} \,.
$$

The first term can be bounded as follows:

$$
\mathbb{E} \sup_{u\in\mathcal{U}_s} g^\mathsf{T}\Sigma^{1/2}u \leqslant \mathbb{E}\|u\|_1 \cdot \|\Sigma^{1/2}g\|_{\max} \leqslant s\,\mathbb{E}\|\Sigma^{1/2}g\|_{\max} \leqslant 2s\sqrt{\nu(\Sigma)\log d} \,,
$$

where we used Fact H.4 to bound the max norm of a vector $\Sigma^{1/2}g$ whose entries are $\nu(\Sigma)$-subgaussian. Dividing by $\sqrt{n}$, we get the desired bound. $\qquad\square$

Now, we bound the deviation of $\mathcal{M}_s(X)$:

**Lemma E.5.** *For all $t \geqslant 0$,*

$$
\mathbb{P}[|\mathcal{M}_s(X) - \mathbb{E}\,\mathcal{M}_s(X)| \geqslant t] \leqslant 2\exp\big[-nt^2/2\big] \,.
$$

*Proof.* For $A \in \mathbb{R}^{n\times d}$ denote $\mathcal{F}_s(A) = \sqrt{n} \cdot \mathcal{M}_s(A\Sigma^{1/2}) = \sup_{u\in\mathcal{U}_s}\|A\Sigma^{1/2}u\|$. Note that for all $A, B \in \mathbb{R}^{n\times d}$,

$$
\mathcal{F}_s(A) - \mathcal{F}_s(B) \leqslant \sup_{u\in\mathcal{U}_s}\big(\|A\Sigma^{1/2}u\| - \|B\Sigma^{1/2}u\|\big) \leqslant \sup_{u\in\mathcal{U}_s}\|(A-B)\Sigma^{1/2}u\| \leqslant \|A - B\| \leqslant \|A - B\|_\mathrm{F} \,.
$$

Hence $\mathcal{F}_s$ is 1-Lipschitz, and by Fact H.5, for all $\tau \geqslant 0$,

$$
\mathbb{P}[|\mathcal{F}_s(G) - \mathbb{E}\,\mathcal{F}_s(G)| \geqslant \tau] \leqslant 2\exp\big[-\tau^2/2\big] \,,
$$

where $G = X\Sigma^{-1/2}$ is a matrix with i.i.d. standard Gaussian entries. Taking $\tau = t\sqrt{n}$, we get the desired bound. $\qquad\square$

Taking $t = 0.2$, we conclude that with probability at least $1 - 2\exp(-0.02n)$,

$$
\mathcal{M}_s(X) \leqslant 1.2 + 2s\sqrt{\frac{\nu(\Sigma)\log d}{n}} \,.
$$

For $s = \sqrt{K/\sigma_{\min}(\Sigma)}$ this bound implies that for all $u$ such that $\|\Sigma^{1/2}u\| = 1$ and $\|u\|_1 \leqslant \sqrt{K}\|u\|$, with probability at least $1 - 2\exp(-0.02n)$,

$$
\frac{1}{\sqrt{n}}\|Xu\| \leqslant 1.2 + 2\sqrt{\frac{\nu(\Sigma)\,K\log d}{\sigma_{\min}(\Sigma)\cdot n}} \leqslant 1.3 \,,
$$

and we get the desired bound. $\qquad\square$

**Well-spreadness of Gaussian matrices.** If $n \gtrsim \frac{v(\Sigma)}{\sigma_{\min}(\Sigma)} \cdot k \log d$, then an $n \times d$ random matrix $X$ with i.i.d. rows $X_1, \dots, X_n \sim N(0, \Sigma)$ satisfies the RE-property with parameter $\sigma_{\min}(\Sigma)/4$ over all sets of size $k$ where $\sigma_{\min}(\Sigma)$ is the smallest eigenvalue of $\Sigma$ (it is a consequence of Theorem E.3). Also, norms of columns of $X$ are bounded by $O\left(\sqrt{v(\Sigma)n}\right)$ with high probability. Hence, $X$ satisfies Assumption 1 and 2 of Theorem D.1, with high probability.

In the next lemma, we show that it also satisfies the last assumption, namely the well-spreadness assumption:

**Lemma E.6.** *Let $X$ be a n-by-d random matrix with i.i.d. rows $X_1, \dots X_n \sim N(0, \Sigma)$, where $\Sigma$ is a positive definite matrix. Suppose that for some $K \geqslant 1$, $n \geqslant 10^6 \cdot \frac{v(\Sigma)}{\sigma_{\min}(\Sigma)} \cdot K \log d$. Then with probability at least $1 - \exp(-n/1000)$, for all $u \in \mathbb{R}^d$ such that $\|u\|_1 \leqslant \sqrt{K}\|u\|$ and for all sets $S \subseteq [n]$ of size $\lceil 0.999n \rceil$,*

$$\|X_S u\| \geqslant \frac{1}{2}\|Xu\| . \tag{E.2}$$

*Proof.* For a set $M \subseteq [n]$ of size at most $n/1000$ independent of $X$, Theorem E.3 implies that $\|X_M u\| \leqslant 0.1\sqrt{n} \cdot \|\Sigma^{1/2}u\|$ and $\|Xu\| \geqslant 0.5\sqrt{n} \cdot \|\Sigma^{1/2}u\|$ with probability at least $1 - 2\exp(-n/100)$. Using a union bound over all sets $M$ of size $n - \lceil 0.999n \rceil$, with probability

$$1 - 2\exp[-n/100 + n\log(1000e)/1000] \geqslant 1 - \exp(-n/1000) ,$$

we get

$$\|X_M u\|^2 \leqslant 0.1\|Xu\|^2 .$$

Since for $S = [n] \setminus M$, $\|Xu\|^2 = \|X_M u\|^2 + \|X_S u\|^2$, we get the desired bound. $\qquad \square$

Now we can prove restricted strong convexity.

**Restricted local strong convexity of Huber loss for Gaussian design.**

**Lemma E.7.** *Consider the settings of Theorem 2.3. Let $0 < \delta < 1, R > 0$. Define*

$$\mathcal{B}_R := \left\{ u \in \mathbb{R}^d \;\middle|\; \frac{1}{\sqrt{n}}\|X(u)\| = R \right\} .$$

*Suppose that $R \leqslant \frac{1}{200}$. Then with probability at least $1 - 3\exp(-\alpha n/1000)$, for all $u \in \mathcal{B}_R \cap \mathcal{S}_4(\Omega)$,*

$$F_2(\beta^* + u) \geqslant F_2(\beta^*) + \langle \nabla F_2(\beta^*), u \rangle + \frac{\alpha n}{200} \cdot \frac{1}{n}\|Xu\|^2 .$$

*Proof.* Let $u \in \mathcal{B}_R \cap \mathcal{S}_4(\Omega)$, where $\Omega$ is the support of $\beta^*$. By Lemma G.2,

$$F_2(\beta^* + u) - F_2(\beta^*) - \langle \nabla F_2(\beta^*), u \rangle \geqslant \frac{1}{2}\sum_{i\in[n]} \langle X_i, u \rangle^2 \mathbf{1}_{[|\eta_i|\leqslant 1]} \cdot \mathbf{1}_{[|\langle X_i, u \rangle|\leqslant 1]} .$$

Denote $A = \left\{ i \in [n] \;\middle|\; |\eta_i| \leqslant 1 \right\}$. Matrix $X_A$ is an $\alpha n \times d$ random matrix with i.i.d. rows $X_j \sim N(0, \Sigma)$. By Theorem E.3, with probability $1 - 2\exp(-\alpha n/100)$,

$$16\alpha R^2 n = 16\alpha\|Xu\|^2 \geqslant 4\alpha n\left\|\Sigma^{1/2}u\right\|^2 \geqslant \|X_A u\|^2 \geqslant \frac{\alpha n}{4}\left\|\Sigma^{1/2}u\right\|^2 \geqslant \frac{\alpha}{16}\|Xu\|^2 = \frac{\alpha}{16}R^2 n .$$

By Lemma E.6, with probability $1 - \exp(-\alpha n/1000)$, $X_A$ satisfies well-spread property for sets of size $\alpha n/1000$ and for all $u \in \mathcal{S}_4(\mathcal{K})$. Since number of entries of $X_A u$ which are larger than 1 is at most $16\alpha R^2 n \leqslant \alpha n/1000$, we get

$$\sum_{i\in[n]} \langle X_i, u \rangle^2 \mathbf{1}_{[|\eta_i|\leqslant 1]} \cdot \mathbf{1}_{[|\langle X_i, u \rangle|\leqslant 1]} = \sum_{i\in A} \langle X_i, u \rangle^2 \mathbf{1}_{[|\langle X_i, u \rangle|\leqslant 1]} \geqslant \frac{1}{4}\|X_A u\|^2 \geqslant \frac{\alpha}{64}R^2 n .$$

Hence with probability at least $1 - 3\exp(-\alpha n/1000)$ we get the desired bound. $\qquad \square$

**Putting everything together.** Let's check that the conditions of Theorem B.1 are satisfied for $\Omega = \overline{\Omega} = \text{supp}(\beta^*)$ and $\mathcal{E}(u) = \frac{1}{\sqrt{n}}\|Xu\|$. Decomposability is obvious. As a consequence of Theorem E.3 $X$ satisfies the RE-property with $\lambda \geqslant \sigma_{\min}(\Sigma)/4$ with probability at least $1 - \exp(-n/100)$, so contraction is satisfied with $s = 8\sqrt{k/\sigma_{\min}(\Sigma)}$. By Lemma E.2, gradient is bounded by $15\sqrt{\nu(\Sigma) \cdot n \cdot (\log d)}$ with probability $1 - d^{-10}/2$. By Lemma E.7, with probability at least $1 - 3\exp(n/1000)$, Huber loss satisfies restricted local strong convexity with parameter $\kappa = 0.01\alpha n$. Hence for

$$n \gtrsim \frac{\nu(\Sigma) \cdot k \log d}{\sigma_{\min}(\Sigma) \cdot \alpha^2} \qquad \text{and} \qquad R \gtrsim \sqrt{\frac{\nu(\Sigma) \cdot k \log d}{\sigma_{\min}(\Sigma) \cdot \alpha^2 \cdot n}}$$

and since then we have $\hat{\beta} - \beta^* \in \mathcal{B}_R \cap \mathcal{S}_4(\Omega)$ the estimator $\hat{\beta}$ defined in Eq. (D.1) satisfies $\frac{1}{\sqrt{n}}\left\|X\left(\hat{\beta} - \beta^*\right)\right\| \leqslant R$ with probability at least $1 - d^{-10}$. Since $\hat{\beta} - \beta \in \mathcal{S}_4(\Omega)$, we also get the desired parameter error $\left\|\hat{\beta} - \beta\right\| \leqslant 2R/\sqrt{\sigma_{\min}(\Sigma)}$.

# F   Optimal fraction of inliers for principal component analysis under oblivious noise (Theorem 2.4)

In this section we prove Theorem 2.4. Recall that a successful $(\varepsilon, \delta)$-weak recovery algorithm (where $\varepsilon, \delta \in (0,1)$) for PCA is an algorithm that takes $Y$ as input and returns a matrix $\hat{L}$ such that $\left\|\hat{L} - L^*\right\|_F \leqslant \varepsilon \cdot \rho$ with probability at least $1 - \delta$ (where $\rho$, $Y$ and $L^*$ are as in Theorem 2.1).

Let's restate Theorem 2.4:

**Theorem F.1** (Restatement of Theorem 2.4). *Let $Y = L^* + N \in \mathbb{R}^{n \times n}$, where $\text{rank}(L^*) = r$, $\|L^*\|_{\max} \leqslant \rho/n$ and the entries of $N$ are independent and symmetric about zero. Let $\zeta \geqslant 0$.*

*Then there exists a universal constant $C_0 > 0$ such that for every $0 < \varepsilon < 1$ and $0 < \delta < 1$, if $\alpha := \min_{i,j \in [n]} \mathbb{P}[|N_{i,j}| \leqslant \zeta]$ satisfies $\alpha < C_0 \cdot (1 - \varepsilon^2)^2 \cdot (1 - \delta) \cdot \sqrt{r/n}$, and $n$ is large enough, then it is information-theoretically impossible to have a successful $(\varepsilon, \delta)$-weak recovery algorithm. The problem remains information-theoretically impossible (for the same regime of parameters) even if we assume that $L^*$ is incoherent; more precisely, even if we know that $L^*$ has incoherence parameters that are as good as those of a random flat matrix of rank $r$, the theorem still holds.*

More in detail, we construct distributions over $L^*$ and $N$ such that the assumptions of the theorem are satisfied and if $\alpha < C_0 \cdot (1 - \varepsilon^2)^2 \cdot (1 - \delta) \cdot \sqrt{r/n}$, weak recovery is not possible.

We will assume without loss of generality that $0 \leqslant \zeta \leqslant \rho/n = 1$. Indeed, weak recovery property is scale invariant, so we can assume $\rho = n$. We can assume $\zeta \leqslant 1$ since if the theorem is true for $\zeta = 1$, then it is true for all $\zeta > 1$.

## A generative model for the hidden matrix

In the following, we will denote the all-zeros vector of dimension $n$ as $\mathbf{0}_n$. Similarly, we will denote the all-ones vector of dimension $n$ as $\mathbf{1}_n$.

For the sake of simplicity, we will assume that $\frac{n}{r}$ is an integer.[13] We will divide the the matrix $L^*$ into $r$ blocks of $\frac{n}{r} \times n$ sub-matrices.

For every $1 \leqslant k \leqslant r$, let $u_k$ be an arbitrary but fixed and deterministic vector in the set $\left\{\mathbf{0}_{(k-1) \cdot \frac{n}{r}}\right\} \times \{-1, +1\}^{\frac{n}{r}} \times \left\{\mathbf{0}_{(r-k) \cdot \frac{n}{r}}\right\}$, and let $\mathbf{v}_k$ be a random flat vector chosen uniformly from $\{-1, +1\}^n$. We further assume that the random vectors $\{\mathbf{v}_k\}_{1 \leqslant k \leqslant r}$ are mutually independent. The hidden matrix $L^*$ is constructed as follows:[14]

---

[13] All the subsequent proofs can be adapted for a general $r$ with minor modifications.

[14] For the general case in which $\frac{n}{r}$ may not be an integer, we divide $L^*$ into $r$ blocks of disjoint sub-matrices of dimensions $\lfloor \frac{n}{r} \rfloor \times n$ and $\lceil \frac{n}{r} \rceil \times n$.

$$L^* = \sum_{k=1}^{r} u_k \cdot \mathbf{v}_k^T.$$

Note that $L^*$ is a flat matrix, i.e., $L^* \in \{-1, +1\}^{n \times n}$. Furthermore, the rank of $L^*$ is at most $r$, and with high probability, $L^*$ is incoherent with parameter $\mu \leq O(\log n)$.

**The noise distribution**

Let $(N_{ij})_{i,j \in [n]}$ be i.i.d. random variables that are sampled according to the distribution

$$\mathbb{P}[N_{i,j} = \ell] = \begin{cases} \dfrac{\xi \sqrt{r}}{2\sqrt{n} - \xi \sqrt{r}} \left(1 - \xi \sqrt{\dfrac{r}{n}}\right)^{|\ell|/2} & \text{if } \ell \in 2\mathbb{Z} = \{\ldots, -4, -2, 0, 2, 4, \ldots\}, \\ \\ 0 & \text{otherwise,} \end{cases} \quad \text{(F.1)}$$

where $0 < \xi \leq 1/2$ is a constant. Furthermore, we assume that $N$ is independent from $L^*$. The distribution of $N$ is symmetric and satisfies

$$\alpha := \mathbb{P}[|N_{ij}| \leq 1] = \mathbb{P}[N_{ij} = 0] = \frac{\xi \sqrt{r}}{2\sqrt{n} - \xi \sqrt{r}} = \Theta\left(\xi \sqrt{\frac{r}{n}}\right).$$

Define

$$Y = L^* + N.$$

**Upper bound on the mutual information**

**Lemma F.2.** *The mutual information $I(L^*; Y)$ between $L^*$ and $Y$ can be upper bounded as follows:*

$$I(L^*; Y) \leq O(\xi \cdot n \cdot r).$$

*Proof.* Notice that for every $\ell \in 2\mathbb{Z}$, we have

$$\mathbb{P}[N_{ij} = \ell + 2] = \mathbb{P}[N_{ij} = \ell] \cdot \left(1 - \xi \sqrt{\frac{r}{n}}\right)^{\text{sign}(\ell+1)}, \quad \text{(F.2)}$$

where

$$\text{sign}(x) = \begin{cases} 1 & \text{if } x > 0, \\ -1 & \text{if } x < 0. \end{cases}$$

For every $L^* \in \{-1, +1\}^{n \times n}$ and every $Y \in (2\mathbb{Z} + 1)^{n \times n}$, we have

$$\begin{aligned} \mathbb{P}[Y = Y | L^* = L^*] &= \mathbb{P}[N = Y - L^*] \\ &= \prod_{i,j} \mathbb{P}[N_{ij} = Y_{ij} - L_{ij}^*] \\ &= \prod_{i,j} \mathbb{P}[N_{ij} = Y_{ij} - 1 + 1 - L_{ij}^*] \\ &\overset{(*)}{=} \prod_{i,j} \left[\mathbb{P}[N_{ij} = Y_{ij} - 1] \cdot \left(1 - \xi\sqrt{\frac{r}{n}}\right)^{\frac{1}{2} \cdot (1 - L_{ij}^*) \cdot \text{sign}(Y_{ij} - 1 + 1)}\right] \\ &= \mathbb{P}[N = Y - \mathbf{1}_n \mathbf{1}_n^T] \cdot \prod_{i,j} \left(1 - \xi\sqrt{\frac{r}{n}}\right)^{\frac{1}{2} \cdot (1 - L_{ij}^*) \cdot \text{sign}(Y_{ij})}, \end{aligned}$$

where (∗) follows from Eq. (F.2). Therefore, we can write

$$\mathbb{P}[\boldsymbol{Y} = Y | \boldsymbol{L}^* = L^*] = \mathbb{P}\big[\boldsymbol{N} = Y - \mathbf{1}_n \mathbf{1}_n^T\big] \cdot f(L^*, Y), \tag{F.3}$$

where

$$
\begin{aligned}
f(L^*, Y) &= \prod_{i,j} \left(1 - \xi\sqrt{\frac{r}{n}}\right)^{\frac{1}{2} \cdot (1 - L_{i,j}^*) \cdot \mathrm{sign}(Y_{ij})} \\
&= \left(1 - \xi\sqrt{\frac{r}{n}}\right)^{\frac{1}{2} \cdot \sum_{i,j} (1 - L_{i,j}^*) \cdot \mathrm{sign}(Y_{ij})} \\
&= \left(1 - \xi\sqrt{\frac{r}{n}}\right)^{\frac{1}{2} \cdot \langle \mathbf{1}_n \mathbf{1}_n^T - L^*, \mathrm{sign}(Y)\rangle},
\end{aligned}
\tag{F.4}
$$

where $\mathrm{sign}(Y)$ is the $n \times n$ matrix defined as $\mathrm{sign}(Y)_{i,j} = \mathrm{sign}(Y_{i,j})$. Furthermore, from Eq. (F.3) we can deduce that for every $Y \in (2\mathbb{Z} + 1)^{n \times n}$, we have

$$\mathbb{P}[\boldsymbol{Y} = Y] = \mathbb{P}\big[\boldsymbol{N} = Y - \mathbf{1}_n \mathbf{1}_n^T\big] \cdot g(Y), \tag{F.5}$$

where

$$g(Y) = \mathbb{E}_{\boldsymbol{L}^*}[f(\boldsymbol{L}^*, Y)]. \tag{F.6}$$

Now from Hölder's inequality, we have

$$
\begin{aligned}
\big|\langle \mathbf{1}_n \mathbf{1}_n^T - L^*, \mathrm{sign}(Y)\rangle\big| &\leq \big\|\mathbf{1}_n \mathbf{1}_n^T - L^*\big\|_{\mathrm{nuc}} \cdot \|\mathrm{sign}(Y)\| \\
&\leq \left(\big\|\mathbf{1}_n \mathbf{1}_n^T\big\|_{\mathrm{nuc}} + \|L^*\|_{\mathrm{nuc}}\right) \cdot \|\mathrm{sign}(Y)\| \\
&= (n + \|L^*\|_{\mathrm{nuc}}) \cdot \|\mathrm{sign}(Y)\|.
\end{aligned}
$$

If $L^*$ is in the support of the distribution of $\boldsymbol{L}^*$, then there exist $r$ vectors $\{v_k\}_{1 \leq k \leq r}$ such that $v_k \in \{-1, +1\}^n$ for every $1 \leq k \leq r$, and

$$L^* = \sum_{k=1}^{r} u_k \cdot v_k^T.$$

Therefore,

$$\|L^*\|_{\mathrm{nuc}} \leq \sum_{k=1}^{r} \big\|u_k \cdot v_k^T\big\|_{\mathrm{nuc}} = \sum_{k=1}^{r} \sqrt{\frac{n}{r}} \cdot \sqrt{n} = r\frac{n}{\sqrt{r}} = n\sqrt{r}. \tag{F.7}$$

Hence,

$$\big|\langle \mathbf{1}_n \mathbf{1}_n^T - L^*, \mathrm{sign}(Y)\rangle\big| \leq n \cdot (\sqrt{r} + 1) \cdot \|\mathrm{sign}(Y)\| \leq 2n\sqrt{r} \cdot \|\mathrm{sign}(Y)\|.$$

By combining this with Eq. (F.4), we get

$$\left(1 - \xi\sqrt{\frac{r}{n}}\right)^{n\sqrt{r} \cdot \|\mathrm{sign}(Y)\|} \leq f(L^*, Y) \leq \left(1 - \xi\sqrt{\frac{r}{n}}\right)^{-n\sqrt{r} \cdot \|\mathrm{sign}(Y)\|}. \tag{F.8}$$

Furthermore, from Eq. (F.6) and Eq. (F.8), we get

$$\left(1 - \xi\sqrt{\frac{r}{n}}\right)^{n\sqrt{r} \cdot \|\mathrm{sign}(Y)\|} \leq g(Y) \leq \left(1 - \xi\sqrt{\frac{r}{n}}\right)^{-n\sqrt{r} \cdot \|\mathrm{sign}(Y)\|}. \tag{F.9}$$

The mutual information between $\boldsymbol{L}^*$ and $\boldsymbol{Y}$ is given by

$$I(\boldsymbol{L}^*; \boldsymbol{Y}) = \sum_{L^*, Y} \mathbb{P}[\boldsymbol{L}^* = L^*, \boldsymbol{Y} = Y] \cdot \log_2 \frac{\mathbb{P}[\boldsymbol{Y} = Y | \boldsymbol{L}^* = L^*]}{\mathbb{P}[\boldsymbol{Y} = Y]}$$

$$\overset{(\dagger)}{=} \sum_{L^*, Y} \mathbb{P}[\boldsymbol{L}^* = L^*, \boldsymbol{Y} = Y] \cdot \log_2 \frac{\mathbb{P}[\boldsymbol{N} = Y - \mathbf{1}_n \mathbf{1}_n^T] \cdot f(L^*, Y)}{\mathbb{P}[\boldsymbol{N} = Y - \mathbf{1}_n \mathbf{1}_n^T] \cdot g(Y)}$$

$$= \sum_{L^*, Y} \mathbb{P}[\boldsymbol{L}^* = L^*, \boldsymbol{Y} = Y] \cdot \log_2 \frac{f(L^*, Y)}{g(Y)}$$

$$= \mathbb{E}\left[\log_2 \frac{f(\boldsymbol{L}^*, \boldsymbol{Y})}{g(\boldsymbol{Y})}\right],$$

where (†) follows from Eq. (F.3) and Eq. (F.5). Now from Eq. (F.8) and Eq. (F.9), we get

$$I(\boldsymbol{L}^*; \boldsymbol{Y}) \leqslant \mathbb{E}\left[\log_2\left(\left(1 - \xi\sqrt{\frac{r}{n}}\right)^{-2n\sqrt{r} \cdot \|\mathrm{sign}(\boldsymbol{Y})\|}\right)\right]$$

$$= -2n\sqrt{r} \cdot \log_2\left(1 - \xi\sqrt{\frac{r}{n}}\right) \cdot \mathbb{E}[\|\mathrm{sign}(\boldsymbol{Y})\|]$$

$$= -\frac{2}{\log 2} n\sqrt{r} \cdot \log\left(1 - \xi\sqrt{\frac{r}{n}}\right) \cdot \mathbb{E}[\|\mathrm{sign}(\boldsymbol{Y})\|] \qquad \text{(F.10)}$$

$$\overset{(\ddagger)}{\leqslant} \frac{4}{\log 2} n\sqrt{r} \cdot \xi\sqrt{\frac{r}{n}} \cdot \mathbb{E}[\|\mathrm{sign}(\boldsymbol{Y})\|]$$

$$= \frac{4\xi \cdot \sqrt{n} \cdot r}{\log 2} \cdot \mathbb{E}[\|\mathrm{sign}(\boldsymbol{Y})\|],$$

where (‡) follows from the fact that $-\log(1 - t) \leqslant 2t$ for every $t \in [0, 1/2]$.

Now let $\boldsymbol{S} = \mathrm{sign}(\boldsymbol{Y})$. In order to estimate $\mathbb{E}[\|\mathrm{sign}(\boldsymbol{Y})\|] = \mathbb{E}[\|\boldsymbol{S}\|]$, we first condition on $\boldsymbol{L}^* = L^*$ for a fixed $L^*$:

$$\mathbb{E}\left[\|\boldsymbol{S}\| \big| \boldsymbol{L}^* = L^*\right] = \mathbb{E}\left[\|\boldsymbol{S} - \mathbb{E}[\boldsymbol{S}|\boldsymbol{L}^* = L^*] + \mathbb{E}[\boldsymbol{S}|\boldsymbol{L}^* = L^*]\| \big| \boldsymbol{L}^* = L^*\right]$$

$$\leqslant \mathbb{E}\left[\|\boldsymbol{S} - \mathbb{E}[\boldsymbol{S}|\boldsymbol{L}^* = L^*]\| \big| \boldsymbol{L}^* = L^*\right] + \|\mathbb{E}[\boldsymbol{S}|\boldsymbol{L}^* = L^*]\|.$$

Notice that

$$\mathbb{E}\left[\boldsymbol{S}\big|\boldsymbol{L}^* = L^*\right] = \mathbb{E}\left[\mathrm{sign}(L^* + \boldsymbol{N})\big|\boldsymbol{L}^* = L^*\right]$$

$$= \mathbb{E}[\mathrm{sign}(L^* + \boldsymbol{N})]$$

$$= \alpha \cdot L^*,$$

where

$$\alpha = \mathbb{P}[\boldsymbol{N}_{ij} = 0] = \frac{\xi\sqrt{r}}{2\sqrt{n} - \xi\sqrt{r}}.$$

Therefore,

$$\mathbb{E}\left[\|\boldsymbol{S}\| \big| \boldsymbol{L}^* = L^*\right] \leqslant \mathbb{E}\left[\|\hat{\boldsymbol{S}}\| \big| \boldsymbol{L}^* = L^*\right] + \alpha \cdot \|L^*\|,$$

where

$$\hat{\boldsymbol{S}} = \boldsymbol{S} - \mathbb{E}[\boldsymbol{S}|\boldsymbol{L}^* = L^*] = \boldsymbol{S} - \alpha \cdot L^*.$$

Now given $\boldsymbol{L}^* = L^*$, the entries of $\hat{\boldsymbol{S}}$ are centered and conditionally mutually independent. Furthermore, $\|\hat{\boldsymbol{S}}\|_{\max} \leqslant \|\boldsymbol{S}\|_{\max} + \alpha \cdot \|L^*\|_{\max} = 1 + \alpha \leqslant 2$. Therefore, by Fact H.7, there is a universal constant $C \geqslant 2$ such that

$$\mathbb{E}\left[\|\hat{\boldsymbol{S}}\| \big| \boldsymbol{L}^* = L^*\right] \leqslant C\sqrt{n}.$$

We conclude that

$$\mathbb{E}[\|\mathrm{sign}(\boldsymbol{Y})\|] = \mathbb{E}[\|\boldsymbol{S}\|] \leqslant C\sqrt{n} + \alpha \cdot \mathbb{E}[\|\boldsymbol{L}^*\|]. \qquad \text{(F.11)}$$

Now notice that $\|L^*\| = U \cdot \mathbf{V}^T$, where $U = [u_1 \ \ldots \ u_r]$ is the $n \times r$ matrix whose columns are $u_1, \ldots, u_r$, and $\mathbf{V} = [\mathbf{v}_1 \ \ldots \ \mathbf{v}_r]$ is the $n \times r$ matrix whose columns are $\mathbf{v}_1, \ldots, \mathbf{v}_r$. We have:

$$\mathbb{E}[\|L^*\|] = \mathbb{E}[\|U \cdot \mathbf{V}^T\|] \leqslant \mathbb{E}[\|U\| \cdot \|\mathbf{V}^T\|] = \|U\| \cdot \mathbb{E}[\|\mathbf{V}^T\|]$$

$$= \sqrt{\frac{n}{r}} \cdot \mathbb{E}[\|\mathbf{V}^T\|] \overset{(\wr)}{\leqslant} \sqrt{\frac{n}{r}} \cdot C\sqrt{n} = C\frac{n}{\sqrt{r}},$$

where $(\wr)$ follows from the fact that $\mathbf{V}$ is an $n \times r$ matrix with i.i.d. zero-mean entries and satisfying $\|\mathbf{V}\|_{\max} = 1$ and Fact H.7. By inserting this in Eq. (F.11), we get

$$\mathbb{E}[\|\mathrm{sign}(Y)\|] \leqslant C\sqrt{n} + \frac{\xi\sqrt{r}}{2\sqrt{n} - \xi\sqrt{r}} \cdot C\frac{n}{\sqrt{r}}$$

$$\leqslant C\sqrt{n} + \frac{\sqrt{r}}{\sqrt{n}} \cdot C\frac{n}{\sqrt{r}}$$

$$= 2C\sqrt{n},$$

By combining this with Eq. (F.10), we get

$$I(L^*; Y) \leqslant \frac{4\xi \cdot \sqrt{n} \cdot r}{\log 2} \cdot 2C\sqrt{n}$$

$$\leqslant \frac{8C\xi}{\log 2} \cdot n \cdot r$$

$$= O(\xi \cdot n \cdot r).$$

$\square$

**Successful weak-recovery reduces entropy**

**Lemma F.3.** *If there exists a $(\delta, \varepsilon)$-successful weak recovery algorithm that takes $Y = L^* + N$ as input and returns a matrix $\hat{L}$ as output in such a way that*

$$\mathbb{P}[\|\hat{L} - L^*\|_{\mathrm{F}} \leqslant \varepsilon \cdot n] \geqslant 1 - \delta,$$

*then the mutual information between $L^*$ and $Y$ can be lower bounded as follows:*

$$I(L^*; Y) \geqslant \frac{(1 - \varepsilon^2)^2}{8 \log 2} \cdot (1 - \delta) \cdot n \cdot r - 1.$$

*Proof.* Define the set

$$\Omega = \left\{ \sum_{k=1}^{r} u_k v_k^T : \ \forall k \in [r], \ v_k \in \mathbb{R}^n \text{ and } \|v_k\|_{\max} \leqslant 1 \right\}.$$

It is easy to see that $\Omega$ is a closed and convex set. Let $\hat{L}_\Omega$ be the orthogonal projection of $\hat{L}$ onto $\Omega$. Since $L^* \in \Omega$, we have $\|\hat{L}_\Omega - L^*\|_{\mathrm{F}} \leqslant \|\hat{L} - L^*\|_{\mathrm{F}}$. Therefore,

$$\mathbb{P}[\|\hat{L}_\Omega - L^*\|_{\mathrm{F}} \leqslant \varepsilon \cdot n] \geqslant \mathbb{P}[\|\hat{L} - L^*\|_{\mathrm{F}} \leqslant \varepsilon \cdot n] \geqslant 1 - \delta.$$

Using an inequality that is similar to the standard Fano-inequality, we will show that the existence of a successful weak-recovery algorithm implies a linear decrease in the entropy of the random vectors $(\mathbf{v}_k)_{k \in [r]}$.

Define the random variable $Z$ as follows:

$$Z = \mathbf{1}_{[\|\hat{L}_\Omega - L^*\|_{\mathrm{F}} \leqslant \varepsilon \cdot n]}.$$

Furthermore, for every $L \in \Omega$, define

$$B_{L,\varepsilon} = \left\{ (v_k)_{k \in [r]} \in \{-1, +1\}^{n \cdot r} : \left\| L - \sum_{k=1}^{r} u_k v_k^{\mathsf{T}} \right\|_{\mathrm{F}} \leqslant \varepsilon \cdot n \right\}.$$

Clearly, if $\mathbf{Z} = 1$, then $(\mathbf{v}_k)_{k \in [r]} \in B_{\hat{L}_\Omega, \varepsilon}$.

Let $H\big((\mathbf{v}_k)_{k \in [r]} \big| \hat{L}_\Omega\big)$ be the conditional entropy of $(\mathbf{v}_k)_{k \in [r]}$ given $\hat{L}_\Omega$. We have:

$$
\begin{aligned}
H\big((\mathbf{v}_k)_{k \in [r]} \big| \hat{L}_\Omega\big) &\leqslant H\big(\mathbf{Z}, (\mathbf{v}_k)_{k \in [r]} \big| \hat{L}_\Omega\big) \\
&= H\big(\mathbf{Z} \big| \hat{L}_\Omega\big) + H\big((\mathbf{v}_k)_{k \in [r]} \big| \hat{L}_\Omega, \mathbf{Z}\big) \\
&\leqslant H(\mathbf{Z}) + H\big((\mathbf{v}_k)_{k \in [r]} \big| \hat{L}_\Omega, \mathbf{Z} = 0\big) \cdot \mathbb{P}[\mathbf{Z} = 0] + H\big((\mathbf{v}_k)_{k \in [r]} \big| \hat{L}_\Omega, \mathbf{Z} = 1\big) \cdot \mathbb{P}[\mathbf{Z} = 1] \\
&\leqslant 1 + n \cdot r \cdot \mathbb{P}[\mathbf{Z} = 0] + (1 - \mathbb{P}[\mathbf{Z} = 0]) \cdot H\big((\mathbf{v}_k)_{k \in [r]} \big| \hat{L}_\Omega, \mathbf{Z} = 1\big),
\end{aligned}
$$

where the last inequality follows from the fact that $\mathbf{Z}$ is a binary random variable (and hence $H(\mathbf{Z}) \leqslant \log_2(2) = 1$), and the fact that $(\mathbf{v}_k)_{k \in [r]} \in \{-1, +1\}^{n \cdot r}$, which implies that $H\big((\mathbf{v}_k)_{k \in [r]} \big| \hat{L}_\Omega, \mathbf{Z} = 0\big) \leqslant \log_2 \big| \{-1, +1\}^{n \cdot r} \big| = n \cdot r$.

Since $\mathbb{P}[\mathbf{Z} = 0] \leqslant \delta$ and $H\big((\mathbf{v}_k)_{k \in [r]} \big| \hat{L}_\Omega, \mathbf{Z} = 1\big) \leqslant \log_2 \big| \{-1, +1\}^{n \cdot r} \big| = n \cdot r$, we have

$$H\big((\mathbf{v}_k)_{k \in [r]} \big| \hat{L}_\Omega\big) \leqslant 1 + n \cdot r + (1 - \delta) \cdot H\big((\mathbf{v}_k)_{k \in [r]} \big| \hat{L}_\Omega, \mathbf{Z} = 1\big).$$

Now notice that

$$H\big((\mathbf{v}_k)_{k \in [r]} \big| \hat{L}_\Omega, \mathbf{Z} = 1\big) \overset{(*)}{\leqslant} \log_2 \big| B_{\hat{L}_\Omega, \varepsilon} \big| \leqslant \max_{L \in \Omega} \log_2 |B_{L,\varepsilon}|,$$

where $(*)$ follows from the fact that given $\mathbf{Z} = 1$, we have $(\mathbf{v}_k)_{k \in [r]} \in B_{\hat{L}_\Omega, \varepsilon}$. On the other hand, for every $L \in \Omega$, we have

$$\log_2 |B_{L,\varepsilon}| = n \cdot r + \log_2 \frac{|B_{L,\varepsilon}|}{2^{n \cdot r}} = n \cdot r + \log_2 \mathbb{P}\big[(\mathbf{v}_k)_{k \in [r]} \in B_{L,\varepsilon}\big],$$

where the last equality follows from the fact that $(\mathbf{v}_k)_{k \in [r]}$ is uniformly distributed in $\{-1, +1\}^{n \cdot r}$. Therefore,

$$H\big((\mathbf{v}_k)_{k \in [r]} \big| \hat{L}_\Omega\big) \leqslant 1 + n \cdot r + (1 - \delta) \cdot \max_{L \in \Omega} \log_2 \mathbb{P}\big[(\mathbf{v}_k)_{k \in [r]} \in B_{L,\varepsilon}\big]. \tag{F.12}$$

Now fix $L \in \Omega$ and let $(v_k)_{k \in [r]}$ be $k$ vectors in $\mathbb{R}^n$ such that $\|v_k\|_{\max} \leqslant 1$ and $L = \sum_{k=1}^{r} u_k v_k^T$. We have $(\mathbf{v}_k)_{k \in [r]} \in B_{L,\varepsilon}$ if and only if $\|L^* - L\|_{\mathrm{F}} \leqslant \varepsilon \cdot n$. Notice that

$$
\begin{aligned}
\|L^* - L\|_{\mathrm{F}}^2 &= \left\langle \sum_{k=1}^{r} u_k \cdot (\mathbf{v}_k - v_k)^T, \sum_{k'=1}^{r} u_{k'} \cdot (\mathbf{v}_{k'} - v_{k'})^T \right\rangle \\
&= \mathrm{Tr}\left( \left( \sum_{k=1}^{r} u_k \cdot (\mathbf{v}_k - v_k)^T \right)^T \cdot \left( \sum_{k'=1}^{r} u_{k'} \cdot (\mathbf{v}_{k'} - v_{k'})^T \right) \right) \\
&= \sum_{k=1}^{r} \sum_{k'=1}^{r} \mathrm{Tr}\big((\mathbf{v}_k - v_k) \cdot u_k^T \cdot u_{k'} \cdot (\mathbf{v}_{k'} - v_{k'})^T\big) \\
&\overset{(\dagger)}{=} \frac{n}{r} \cdot \sum_{k=1}^{r} \mathrm{Tr}\big((\mathbf{v}_k - v_k) \cdot (\mathbf{v}_k - v_k)^T\big) = \frac{n}{r} \cdot \sum_{k=1}^{r} \|\mathbf{v}_k - v_k\|^2 \\
&= \frac{n}{r} \cdot \sum_{k=1}^{r} \sum_{i=1}^{n} (\mathbf{v}_{k,i} - v_{k,i})^2 = \frac{n}{r} \cdot \sum_{k=1}^{r} \sum_{i=1}^{n} \big(\mathbf{v}_{k,i}^2 + v_{k,i}^2 - 2v_{k,i} \cdot \mathbf{v}_{k,i}\big)
\end{aligned}
$$

$$\overset{(\ddagger)}{\geqslant} \frac{n}{r} \cdot \left( n \cdot r - 2 \sum_{k=1}^{r} \sum_{i=1}^{n} v_{k,i} \cdot \mathbf{v}_{k,i} \right),$$

where (†) follows from the fact that $(u_k)_{k \in [r]}$ are orthogonal to each other, and $\|u_k\|^2 = \frac{n}{r}$ for every $k \in [r]$. Note that $(\mathbf{v}_{k,i})_{1 \leqslant i \leqslant n}$ and $(v_{k,i})_{1 \leqslant i \leqslant n}$ are the entries of $\mathbf{v}_k$ and $v_k$, respectively. (‡) follows from the fact that $\mathbf{v}_k \in \{-1, +1\}^n$ for every $k \in [r]$. Therefore,

$$\|L^* - L\|_{\mathrm{F}}^2 \geqslant n^2 - \frac{2n}{r} \cdot \sum_{k=1}^{r} \sum_{i=1}^{n} v_{k,i} \cdot \mathbf{v}_{k,i},$$

which implies that

$$\begin{aligned}
\mathbb{P}\big[(\mathbf{v}_k)_{k \in [r]} \in B_{L,\varepsilon}\big] &= \mathbb{P}\big[\|L^* - L\|_{\mathrm{F}}^2 \leqslant \varepsilon^2 \cdot n^2\big] \\
&\leqslant \mathbb{P}\left[ n^2 - \frac{2n}{r} \cdot \sum_{k=1}^{r} \sum_{i=1}^{n} v_{k,i} \cdot \mathbf{v}_{k,i} \leqslant \varepsilon^2 \cdot n^2 \right] \\
&= \mathbb{P}\left[ \sum_{k=1}^{r} \sum_{i=1}^{n} v_{k,i} \cdot \mathbf{v}_{k,i} \geqslant \frac{1}{2} \cdot (1 - \varepsilon^2) \cdot n \cdot r \right].
\end{aligned}$$

Note that the random variables $(v_{k,i} \cdot \mathbf{v}_{k,i})_{k \in [r], i \in [n]}$ are independent. Moreover, for every $1 \leqslant k \leqslant r$ and every $1 \leqslant i \leqslant n$, we have

$$\mathbb{E}[v_{k,i} \cdot \mathbf{v}_{k,i}] = 0.$$

Furthermore, since $\|v_k\|_{\max} \leqslant 1$ and $\mathbf{v}_k \in \{-1, +1\}^n$, the random variables $(v_{k,i} \cdot \mathbf{v}_{k,i})_{k \in [r], i \in [n]}$ can be uniformly bounded as

$$|v_{k,i} \cdot \mathbf{v}_{k,i}| \leqslant |v_{k,i}| \leqslant 1.$$

It follows from Hoeffding's inequality Fact H.2 that

$$\begin{aligned}
\mathbb{P}\big[(\mathbf{v}_k)_{k \in [r]} \in B_{L,\varepsilon}\big] &\leqslant \exp\left( -\frac{(1 - \varepsilon^2)^2 \cdot n^2 \cdot r^2}{8 \cdot n \cdot r} \right) \\
&= \exp\left( -\frac{(1 - \varepsilon^2)^2}{8} \cdot n \cdot r \right).
\end{aligned}$$

Since this is true for every $L \in \Omega$, we get from Eq. (F.12) that

$$H\big((\mathbf{v}_k)_{k \in [r]} \big| \hat{L}_\Omega\big) \leqslant 1 + n \cdot r - (1 - \delta) \cdot \frac{(1 - \varepsilon^2)^2}{8 \log 2} \cdot n \cdot r.$$

Therefore, the mutual information between $(\mathbf{v}_k)_{k \in [r]}$ and $\hat{L}_\Omega$ satisfies:

$$\begin{aligned}
I\big((\mathbf{v}_k)_{k \in [r]}; \hat{L}_\Omega\big) &= H\big((\mathbf{v}_k)_{k \in [r]}\big) - H\big((\mathbf{v}_k)_{k \in [r]} \big| \hat{L}_\Omega\big) \\
&\geqslant n \cdot r - 1 - n \cdot r + (1 - \delta) \cdot \frac{(1 - \varepsilon^2)^2}{8 \log 2} \cdot n \cdot r \\
&= \frac{(1 - \varepsilon^2)^2}{8 \log 2} \cdot (1 - \delta) \cdot n \cdot r - 1.
\end{aligned}$$

Now since $(\mathbf{v}_k)_{k \in [r]} \to L^* \to Y \to \hat{L} \to \hat{L}_\Omega$ is a Markov chain, it follows from the data-processing inequality that

$$I(L^*; Y) = I\big((\mathbf{v}_k)_{k \in [r]}; Y\big) \geqslant I\big((\mathbf{v}_k)_{k \in [r]}; \hat{L}_\Omega\big) \geqslant \frac{(1 - \varepsilon^2)^2}{8 \log 2} \cdot (1 - \delta) \cdot n \cdot r - 1.$$

$\square$

**Putting everything together**

Now we are ready to prove Theorem 2.4:

*Proof of Theorem 2.4.* From Lemma F.2 and Lemma F.3 we can deduce that if there exists a $(\delta, \varepsilon)$-successful weak recovery algorithm then we must have

$$\frac{8C\xi}{\log 2} \cdot n \cdot r \geq I(L^*; Y) \geq \frac{(1 - \varepsilon^2)^2}{8 \log 2} \cdot (1 - \delta) \cdot n \cdot r - 1.$$

Therefore, if $n$ is large enough and

$$\xi < \frac{(1 - \varepsilon^2)^2}{64C} \cdot (1 - \delta) - \frac{\log 2}{8C} \cdot \frac{1}{r \cdot n},$$

it is impossible to have a $(\delta, \varepsilon)$-successful weak recovery algorithm. Now since $\alpha = \Theta\left(\xi \sqrt{\dfrac{r}{n}}\right)$, we get the result. $\square$

## G   Facts about Huber loss

**Fact G.1** (Integration by parts for absolutely continuous functions). *Let $F, G : \mathbb{R} \to \mathbb{R}$ be absolutely continuous functions, i.e. there exist locally integrable functions $f, g : \mathbb{R} \to \mathbb{R}$ such that for all $a, b \in \mathbb{R}$,*

$$\int_a^b f(t)\,\mathrm{d}t = F(b) - F(a) \qquad \text{and} \qquad \int_a^b g(t)\,\mathrm{d}t = G(b) - G(a).$$

*Then for all $a, b \in \mathbb{R}$,*

$$\int_a^b f(t)G(t)\,\mathrm{d}t = F(b)G(b) - F(a)G(a) - \int_a^b F(t)g(t)\,\mathrm{d}t.$$

*Proof.*

$$
\begin{aligned}
\int_a^b f(t)G(t)\,\mathrm{d}t &= G(a) \cdot (F(b) - F(a)) + \int_a^b f(t) \int_a^b \mathbf{1}_{[\tau \in [a,t]]} g(\tau)\,\mathrm{d}\tau\,\mathrm{d}t && \text{(By definition of } G) \\
&= G(a) \cdot (F(b) - F(a)) + \int_a^b g(\tau) \int_a^b f(t)\mathbf{1}_{[t \in [\tau,b]]}\,\mathrm{d}t\,\mathrm{d}\tau && \text{(By Fubini's theorem)} \\
&= G(a) \cdot (F(b) - F(a)) + \int_a^b g(\tau) \cdot (F(b) - F(\tau))\,\mathrm{d}\tau && \text{(By definition of } F) \\
&= G(a) \cdot (F(b) - F(a)) + F(b)(G(b) - G(a)) - \int_a^b g(\tau)F(\tau)\,\mathrm{d}\tau && \\
&= F(b)G(b) - F(a)G(a) - \int_a^b F(t)g(t)\,\mathrm{d}t. &&
\end{aligned}
$$

$\square$

**Lemma G.2** (Second order behavior of Huber-loss function). *Let $h > 0$. For all $\eta, \delta \in \mathbb{R}$, and all $0 \leq \tau \leq h$,*

$$f_h(\eta + \delta) - f_h(\eta) - f_h'(\eta) \cdot \delta \geq \frac{\delta^2}{2} \mathbf{1}_{[|\eta| \leq h - \tau]} \cdot \mathbf{1}_{[|\delta| \leq \tau]}.$$

*Proof.* Consider $g : \mathbb{R} \to \mathbb{R}$ defined as $g(t) = f_h(\eta + t \cdot \delta)$. Note that for all $a, b \in \mathbb{R}$,

$$f_h'(\eta + b\delta) - f_h'(\eta + a\delta) = \int_{\eta + a\delta}^{\eta + b\delta} \mathbf{1}_{[|x| \leq h]}\,\mathrm{d}x.$$

Changing the variable $x = \eta + t\delta$, we get

$$g'(b) - g'(a) = \delta^2 \int_a^b \mathbf{1}_{[|\eta + t\delta| \leqslant h]} \, \mathrm{d}t \,.$$

By Fact G.1,

$$\delta^2 \int_0^1 \mathbf{1}_{[|\eta + t\delta| \leqslant h]} \cdot (1 - t) \, \mathrm{d}t = -g'(0) + g(1) - g(0) \,.$$

Note that $g(0) = f_h(\eta)$, $g(1) = f_h(\eta + \delta)$ and $g'(0) = \delta f'_h(\eta)$. Since for all $0 \leqslant \tau \leqslant h$, $\mathbf{1}_{[|\eta + t\delta| \leqslant h]} \geqslant \mathbf{1}_{[|\eta| \leqslant h - \tau]} \cdot \mathbf{1}_{[|\delta| \leqslant \tau]}$ and $\int_0^1 (1 - t) \, \mathrm{d}t = 1/2$, we get the desired bound. $\qquad\square$

## H  Tools for Probabilistic Analysis

This section contains some technical results needed for the proofs in the main body of the paper.

**Fact H.1** (Chernoff's inequality, (Ver18))**.** *Let $\zeta_1, \ldots, \zeta_n$ be independent Bernoulli random variables such that $\mathbb{P}(\zeta_i = 1) = \mathbb{P}(\zeta_i = 0) = p$. Then for every $\Delta > 0$,*

$$\mathbb{P}\left(\sum_{i=1}^n \zeta_i \geqslant pn(1 + \Delta)\right) \leqslant \left(\frac{e^{-\Delta}}{(1 + \Delta)^{1 + \Delta}}\right)^{pn} \,.$$

*and for every $\Delta \in (0, 1)$,*

$$\mathbb{P}\left(\sum_{i=1}^n \zeta_i \leqslant pn(1 - \Delta)\right) \leqslant \left(\frac{e^{-\Delta}}{(1 - \Delta)^{1 - \Delta}}\right)^{pn} \,.$$

**Fact H.2** (Hoeffding's inequality, (Wai19))**.** *Let $z_1, \ldots, z_n$ be mutually independent random variables such that for each $i \in [n]$, $z_i$ is supported on $[-c_i, c_i]$ for some $c_i \geqslant 0$. Then for all $t \geqslant 0$,*

$$\mathbb{P}\left(\sum_{i=1}^n (z_i - \mathbb{E}\, z_i) \geqslant t\right) \leqslant \exp\left(-\frac{t^2}{2 \sum_{i=1}^n c_i^2}\right),$$

*and*

$$\mathbb{P}\left(\left|\sum_{i=1}^n (z_i - \mathbb{E}\, z_i)\right| \geqslant t\right) \leqslant 2 \exp\left(-\frac{t^2}{2 \sum_{i=1}^n c_i^2}\right).$$

**Fact H.3** (Bernstein's inequality, (Wai19))**.** *Let $z_1, \ldots, z_n$ be mutually independent random variables such that for each $i \in [n]$, $z_i$ is supported on $[-B, B]$ for some $B \geqslant 0$. Then for all $t \geqslant 0$,*

$$\mathbb{P}\left(\sum_{i=1}^n (z_i - \mathbb{E}\, z_i) \geqslant t\right) \leqslant \exp\left(-\frac{t^2}{2 \sum_{i=1}^n \mathbb{E}\, z_i^2 + \frac{2Bt}{3}}\right).$$

**Fact H.4** (Subgaussian maxima, (Wai19))**.** *Let $d \geqslant 2$ be an integer and let $z$ be a $d$-dimensional random vector with zero mean $\sigma$-subgaussian entires. Then*

$$\mathbb{E}\|z\|_{\max} \leqslant 2\sigma \sqrt{\log d} \,.$$

**Fact H.5** (Lipschitz functions of Gaussian vectors, (Wai19))**.** *Let $g \sim N(0, 1)^m$ for some $m \in \mathbb{N}$ and let $F : \mathbb{R}^m \to \mathbb{R}$ be $L$-Lipschitz with respect to Euclidean norm, where $L > 0$. Then for all $t \geqslant 0$,*

$$\mathbb{P}[|F(g) - \mathbb{E}\, F(g)| \geqslant t] \leqslant 2 \exp\left(-\frac{t^2}{2L^2}\right).$$

**Fact H.6** (Spectral norm of Gaussian matrices, (Wai19))**.** *Let $W \sim N(0, 1)^{n \times d}$. Then*

$$\mathbb{E}\|W\| \leqslant \sqrt{n} + \sqrt{d} \,.$$

*Moreover, for all $t \geqslant 0$,*

$$\mathbb{P}\left[\|W\| \geqslant \sqrt{n} + \sqrt{d} + t\right] \leqslant 2 \exp(-t^2/2) \,.$$

**Fact H.7** (Spectral norm of matrices with bounded independent zero-mean entries, (RV10)). *Let $M$ be an $n$-by-$n$ random matrix with independent zero-mean entries $M_{ij}$ supported on $[-1, 1]$. Then*

$$\mathbb{E}\|M\| \leqslant (2 + o(1))\sqrt{n}$$

*as $n \to \infty$. Moreover, for all $t \geqslant 0$,*

$$\mathbb{P}\left[\|M\| \geqslant \mathbb{E}\|M\| + \sqrt{2\pi} + t\right] \leqslant 2\exp(-t^2/2).$$

**Fact H.8** (Sudakov–Fernique theorem, (Adl90)). *Let $\Theta$ be a compact subset of $\mathbb{R}^m$, where $m \in \mathbb{N}$. Let $W_\theta$ and $Z_\theta$ be real-valued sample-continuous zero-mean Gaussian processes indexed by elements of $\Theta$. Suppose that $\forall \theta, \theta' \in \Theta$, $\mathbb{E}(W_\theta - W_{\theta'})^2 \leqslant \mathbb{E}(Z_\theta - Z_{\theta'})^2$. Then*

$$\mathbb{E}\sup_{\theta \in \Theta} W_\theta \leqslant \mathbb{E}\sup_{\theta \in \Theta} Z_\theta.$$

**Fact H.9** (Sudakov Minoration, (Wai19)). *Let $\{g_\theta \mid \theta \in \Theta\}$ be a zero-mean Gaussian process indexed by elements of some non-empty set $\Theta$. Let $\rho : \Theta \times \Theta \to [0, \infty)$ be a (pseudo)metric $\rho(\theta, \theta') := \left(\mathbb{E}(g_\theta - g_{\theta'})^2\right)^{1/2}$. Then*

$$\mathbb{E}\sup_{\theta \in \mathcal{T}} g_\theta \geqslant \sup_{\varepsilon > 0} \frac{\varepsilon}{2}\sqrt{\log\left|\mathcal{N}_{\varepsilon,\rho}(\Theta)\right|},$$

*where $\left|\mathcal{N}_{\varepsilon,\rho}(\Theta)\right|$ is the minimal size of $\varepsilon$-net in $\Theta$ with respect to $\rho$.*