# OpenReview forum: "Consistent Estimation for PCA and Sparse Regression with Oblivious Outliers"
_NeurIPS.cc/2021/Conference — NeurIPS 2021 Poster_

### Official Review · Reviewer_3eeA · 2021-07-10

**Rating:** 7
**Confidence:** 4

**Summary:**

The authors have proved non-trivial estimation error bounds for one type of robust estimate for sparse linear regression and PCA. Specifically, the estimate is constructed by minimizing the Huber Loss function with L1 regularization.

**Limitations And Societal Impact:**

It is a theoretical paper and the main limitation is that the paper mainly looks at two specific problems.

**Main Review:**

I think overall the paper is well written. For each problem analyzed in this paper, there are sufficient comparisons with the prior work in terms of the sharpness of the error bound and shows when the error bounds derived in this paper are tighter. The discussion of the techniques looks clear. My major concerns of the paper are:
1) While the authors have claimed that their techniques provide a general framework that proves tight error bound for Huber loss-based robust estimate. However, besides the proof details specific to the two problems studied in the paper, I do not find anything new in the approach. To claim a general framework, I am expecting a statement that as long as the problem satisfies certain general properties, then there is a tight error bound for a type of robust estimate.
2) Related to the above point, it is unclear to me why the authors only looked at the Huber loss function. It would be nice and more general to say that as long as the loss function satisfies some assumptions, the error bounds hold. Then point out that Huber loss is one example and hopefully the authors can show a second example of the robust loss function.

Overall, despite the concerns, I think it is a good paper to be accepted.

**Time Spent Reviewing:**

6

---

> ### Author Response · Authors · 2021-08-10
> **Reply to Reviewer 3eeA**
>
> Many thanks for your valuable comments.
>
> While the two problems that we considered are different, we used one meta-theorem (Theorem B.1) to solve both problems. Perhaps we did not emphasize this in the main manuscript, but this is what we meant by a general framework: Both problems can be reduced to an application of one meta-theorem (Theorem B.1). We will emphasize this in the final version. Note that we do not claim that Theorem B.1 is our main contribution. We think that our main contribution lies in the way some estimation problems (where the noise models allow for a large, potentially overwhelming, fraction of oblivious outliers, each of which can have arbitrarily large values) reduce to Theorem B.1. More precisely, our main contribution lies in choosing the correct combination of loss functions and regularizers and, more importantly, in applying the correct analysis to prove optimal performance and thus improving over previous results.
>
> Regarding the loss function, one important aspect of our work is that we did not want to put any constraint on the outliers besides having a symmetric distribution for the noise. This means that the magnitude of the outliers can be arbitrarily large. This somehow forces the loss function to be in such a way that it is linear after some threshold $h$. We still have some freedom in the choice of the loss function in the interval $[-h,h]$: As long as we pick a convex function that gives rise to the strong convexity requirement that we need, everything should work out fine. For example, for arbitrary constant $c\in (1,2]$ one can use the loss function $\sum_{i=1}^n g_c(y_i - \langle X_i, \beta\rangle)$, where $g_c$ is a differentiable function such that for $|x|\le 2$, $g_c(x) = |x|^c$ and for $x<-2$ and $x>2$, $g_c$ is linear. Nevertheless, since the function that is quadratic on the interval is the simplest example of locally strongly convex function, making the loss function to be quadratic in $[-h,h]$ (the interval that should contain the inliers) seems to be the most natural choice. Furthermore, the quadratic function is simple and easily computable, so the optimization algorithm for Huber loss is simple. In summary, there are other loss functions that can work, but the Huber loss seems to be the most natural choice and induces a simple algorithm.

---

### Official Review · Reviewer_y31x · 2021-07-14

**Rating:** 7
**Confidence:** 2

**Summary:**

This paper studies sparse regression and principal component analysis under
oblivious perturbations.

Main contributions:
1. The authors propose estimators that achieve the optimal error guarantees for PCA and sparse regression under oblivious perturbations separately, by minimizing the Huber loss function with some regularization. For sparse regression the estimator also achieves the optimal sample complexity.
2. This machinery has the potential to be applied to other estimation problems.

Post rebuttal: I acknowledge that I have read the authors responses and other reviewers' comments.

**Main Review:**

This paper is reasonably well-written, with clear introduction of the problem and comparison with the related work. The result of the paper improves over the state-of-the-art. Some comments on improving the paper:

1. Current version of the paper is very theoretical, it will be better if some experimental results could be added.
2. The well-spreadness property at line 181 could is not clear: in the current statement it sounds to me like this property is satisfied as long as it holds for any arbitrary choice of $m$. But according to my understanding you do need the $m$ here to have some lower bound (otherwise setting $m = 0$ is trivial). What is the desired size of $m$?
3. Typos in the paper: (1) At line 57, "preturbations" should be "perturbations".

**Time Spent Reviewing:**

6

---

> ### Author Response · Authors · 2021-08-10
> **Reply to Reviewer y31x**
>
>
>
> Many thanks for the review and the suggestions.
>
> We will consider doing experiments in the future.
>
> Regarding the well spreadness property, we need this property for some (large enough) $m$. Concretely, the bound is $m\gtrsim (k\log d)/ (\lambda\cdot \alpha^2)$ (as in Theorem 2.2, line 187). We will clarify this in the final version.
>
> Thank you for pointing out the typo, we will correct it in the final version.

---

### Official Review · Reviewer_ALou · 2021-07-16

**Rating:** 8
**Confidence:** 4

**Summary:**

The authors study PCA and sparse regression in the presence of an oblivious adversary. The observation model is as follows: the observations may have symmetric, independent, additive noise. Each noise term is bounded by $\zeta$ with probability at least $\alpha$ which can be much smaller than 1. The authors show that the estimators which minimize Huber loss with appropriate $\ell_1$ penalty achieve optimal error rates.



**Ethical Concerns:**

No concerns.

**Limitations And Societal Impact:**

No concerns.

**Main Review:**

The authors explain the implications of their results in detail. They give intuition to setup sketch their proofs. Assumptions about the design matrix and the noise are weaker than those in related prior work.

Line 205: \alpha should be \alpha^2
Line 208: use appropriate placeholder constants in front of = and <=


**Time Spent Reviewing:**

5

---

> ### Author Response · Authors · 2021-08-10
> **Reply to Reviewer ALou**
>
> Many thanks for the comments and the evaluation.
>
> We will correct the mentioned typo and address the placeholder constants in the final version.

---

### Official Review · Reviewer_Aoiu · 2021-07-17

**Rating:** 7
**Confidence:** 4

**Summary:**

The paper studies algorithms for sparse regression and principle component analysis using Huber loss in the presence of overwhelming oblivious noise.
In particular, the fraction of inliers, $\alpha$, is going to $0$ but it is still possible to achieve consistency as the noise model is benign (oblivious, symmetric noise).
The paper obtains optimal error rates and dependence on $\alpha$ for a wide range of distribution assumptions.
These rates are achieved by using regularized Huber loss (nuclear norm for the matrix case and $\ell_1$-norm for regression case).



**Limitations And Societal Impact:**

Yes

**Main Review:**

I think the paper makes significant contributions for sparse (and rank-constrained) models in the presence of overwhelming noise. The proof strategy follows the standard analysis of regularized high-dimensional models [Wai19], in combination with a recent line of work on Huber regression for optimal error in robust regression ([SFZ19, PJL20, SF20, dNS21]). The paper is also well-written and usually clear.

I thus recommend "accept". Some comments and suggestions are attached below:

**Main Comment**


+ Citations to several existing works on robust linear linear regression that use Huber loss are missing: [SFZ19, PJL20, SF20]. (Surprisingly [SFZ19] is listed in references but not mentioned in the paper. [SFZ19] is not even cited in the full paper attached as the supplementary material.)  These papers also follow the same proof strategy. In particular, regarding Line 201: The (non-sparse version of) well-spreadness condition looks very similar to the weak stability of [PJL20] (also see [PJL20, Theorem 3.1] ). Although the aforementioned works are looking at a different corruption model, there seems to be a strong relation in techniques, and thus should be discussed.

**Other Comments**

+ Please make it explicit early on that the regression setting only allows corruption in responses (and not the covariates). Assuming clean covariates simplifies the problem considerably.

+ Lines 48-55: It says that the error will be mentioned in $\|\cdot\|$ but the following text in Section $1$ mentions rates for error in $\|\cdot\|^2$.

+ Footnote 3: The second sentence should make it explicit that the reduction is for Gaussian design. I found it confusing at first.

+ Line 123: Define maximum norm. It is less standard than the others.

+ Lines 204-209: it is not clear from these lines that the condition on $\alpha$ is optimal. Please add more details for clarity.

+ When referencing to books, please mention the specific result being used (or a chapter for a broad topic).

**References**

+ [SF20] Sasai, T. & Fujisawa, H. Robust estimation with Lasso when outputs are adversarially contaminated. arXiv:2004.05990 (2020).
+ [PJL20] Pensia, A., Jog, V. & Loh, P. Robust regression with covariate filtering: Heavy tails and adversarial contamination. arXiv:2009.12976 (2020).
+ [SFZ19] Sun, Q., Zhou, W. & Fan, J. Adaptive Huber Regression. arXiv:1706.06991 (2019).

---
After the Author Feedback

I thank the authors for their thoughtful response. I am maintaining my score and thus recommend acceptance of the paper.

**Time Spent Reviewing:**

5

---

> ### Author Response · Authors · 2021-08-10
> **Reply to Reviewer Aoiu**
>
> Many thanks for pointing out the missing references. We will include and discuss them in the final version. In particular, we will include the following:
>
> *Comparison with other works using the estimators based on the Huber Loss*
>
> [SF20] consider sparse parameter vectors but deterministic noise and their analysis is tailored toward this setting. The main difference between our analysis and the one presented in [SFZ19] (which also considers the sparse setting, but requires some moment bounds on the noise entries) is that they choose a Huber Parameter such that most of the outliers lie in the quadratic part, while we choose it in such a way that most lie in the linear parts. [PJL20] study non-sparse linear regression (with the same outlier model, but allowing only a small fraction of unbounded outliers) and also work with the Huber Parameter such that most of the outliers lie in the quadratic part.
>
> It is not clear how the analysis of [SFZ19] (for sparse linear regression) can be applied in our setting. For example, if the design matrix is standard Gaussian, their analysis can only work if the fraction of outliers $1-\alpha$ is less than $1/k$, while our analysis works for $\alpha = o(1)$.
> We will add the detailed comparison to the final version of the paper.
>
> *Comparison with the weak stability property*
>
> The non-sparse version of our well spreadness property is indeed almost the same as the weak stability property in [PJL20]. This notion doesn't work in the sparse setting we consider: it requires a lower bound on the minimum eigenvalue of the empirical covariance matrix which trivially is 0 in the high dimensional case ($n \ll d$). A modification of the weak stability property that would be useful in the sparse case should be similar to the well spreadness property that we use.
>
> Many thanks also for the minor comments, we will address them in the final version. Regarding the optimal dependence on $\alpha$, note that there was a typo in the submitted version: The consistency condition should be $\alpha = \omega\left(\sqrt{\frac{k\log d}{n}}\right)$ instead of $\alpha = \omega\left(\frac{k\log d}{n}\right)$. Note that the Gaussian noise ${\\eta} \sim N(0,\sigma \cdot \text{Id}_n)$ can be seen as a particular case of our model for $\alpha = \mathbb{P}[|\ \eta|\le 1] = \Theta(1/\sigma)$. It is known that for Gaussian noise, consistency can be achieved only if $n=\omega(\sigma^2k\log d)$ (assuming $k \leq d^{1-\Omega(1)}$). This means that in our (more general) model, consistency cannot be achieved if we do not have $n=\omega(\frac{k\log d}{\alpha^2})$, which is equivalent to $\alpha = \omega(\sqrt{\frac{k\log d}{n}})$.

---

> > ### Comment · Reviewer_Aoiu · 2021-08-11
> > **Further questions**
> >
> > I thank the authors for their thoughtful response. I have some follow-up remarks and a short question below:
> >
> > 1. "[PJL20] ... (with *the same outlier model* ....outliers)"
> >
> > [PJL20] actually works under a stronger contamination model than [SF20, SFZ19, dNS21] as it also allows corruption in covariates. Yes, I agree that the fraction of corruption is at most a constant.
> >
> > 2. "[SFZ19].... their analysis can only work *if the fraction of outliers is less than $1/k$*"
> >
> > Can the authors please explain how why this is the case? This claim is not apparent to me from their Theorem 3 (arxiv version). I agree that the goal in that paper is to handle heavy-tails of the response but I believe the underlying proof techniques are similar.
> >
> > 3.*Optimal dependence on $\alpha$*
> >
> > Thanks for clarifying. Perhaps this can be added as a remark environment, and also referred after mentioning Theorem 2.3 (because the tight rate is achieved for the Gaussian case, and it is not clear from Theorem 2.2 how $m,k,d,n$ are related for the Gaussian case.)

---

> > > ### Author Response · Authors · 2021-08-20
> > > **Reply to the follow-up questions of Reviewer Aoiu**
> > >
> > > Thank you very much for the reply.
> > >
> > > Regarding 1:
> > > We agree that the contamination model that allows for adversarially corrupted covariates is significantly stronger (which is also reflected by a significantly larger estimation error for this model that only vanishes if the fraction of covariate corruptions vanishes).
> > > When we update the discussion of related works in our paper, we will emphasize that [PFL20] also considers this stronger contamination model.
> > >
> > > Regarding 2:
> > > We agree that this limitation is not evident from the theorem statements in [SFZ19] as they consider a different model for the oblivious noise. Our comment (which admittedly should have been qualified) was referring to the particular strategy in [SFZ19] for showing strong convexity of the Huber loss function. Concretely, they show that the Huber loss satisfies a strong convexity lower bound up to an additional error term of the form
> > > $\max_{1 \leq i \leq n} \langle u, x_i \rangle^2 \cdot \frac{1}{n} \sum_{i=1}^n 1( | \varepsilon_i | > \tau/2)$
> > > (See Eq (54) in the proof of Lemma 1.)
> > > Here, $x_1,\ldots, x_n$ are the covariates and $\varepsilon_1,\ldots,\varepsilon_n$ the noise variables.
> > > As part of the proof in [SFZ19], one needs to upper bound this term uniformly over all $k$-sparse unit vectors $u$ by a small constant.
> > > Even for standard Gaussian covariates, there are $k$-sparse $u$ with $\max_{1 \leq i \leq n} \langle u, x_i \rangle^2 \ge \Omega(k)$ with high probability.
> > > Hence, one would need $\frac{1}{n} \sum_{i=1}^n 1( | \varepsilon_i | > \tau/2) \le O(1/k)$, which holds under suitable moment bounds for the noise variables but may be violated even for a small ($O(1/k)$) fraction of unbounded outlier noise values.
> > >
> > > Regarding 3:
> > > Thank you for the suggestion. We will implement it in the updated version of our paper.

---

### Decision · Program_Chairs · 2021-09-27

**Decision:**

Accept (Poster)

**Comment:**

This paper studies the fundamental problems of Principal Component Analysis (PCA) and Sparse Linear Regression (SLR) in the presence of oblivious outliers. In the oblivious noise model, a fraction of the labels can be adversarially corrupted and the corruptions are assumed to be independent of the covariates. This corruption model is significantly weaker than other models studied in the recent literature. In particular, recovery is information-theoretically possible even when the fraction of inliers is close to zero. Prior work had developed efficient algorithms for linear regression in the oblivious noise model. The current work extends these previous ideas to PCA and SLR, obtaining efficient algorithms with near-optimal statistical guarantees under a range of distributional assumptions. The reviewers agreed that the contributions are significant and the paper should appear in NeurIPS.